# Photonic spin-Hall effect in chiral plasmonic assemblies

Yilin Chen[1,2,4], Yang Chen[1,4], Yini Fang[1], Ruoqi Ai[1,3], Ximin Cui [3], Xiaolu Zhuo [2] ✉ & Jianfang Wang [1] ✉

Directional light splitting based on the photonic spin-Hall effect is a desired feature for the development of spin-dependent optical elements. Here, we report on the routing of surface plasmon polaritons (SPPs) by using chiral gold nanocubes (Au NCs) and silver nanowires (Ag NWs). We experimentally and theoretically observe the photonic spin-Hall effect in Ag NWs under circularly polarized excitation and show that, when Au NCs of opposite chirality are attached to the NWs, linearly polarized illumination can modulate the SPPs formed on the NWs. We achieve directional emission of valley-polarized excitons from hybrid structures made of (chiral Au NC)–(Ag NW) assembled with transition metal dichalcogenide monolayers and observe an enhanced degree of valley polarization. The underlying mechanism of the routing effect is understood through numerical simulations, confirming that the observed chirality-dependent routing effect can have important implications for the development of valleytronic circuits.

The photonic spin-Hall effect is an optical phenomenon that demonstrates a correlation between the polarization of incident circularly polarized light (CPL) and the direction of excited evanescent waves[1-3]. One interesting result of this effect is the generation of unidirectional evanescent waves propagating in opposite directions with CPL of different handednesses. Spin-controlled directional light splitting has been achieved in both achiral plasmonic and dielectric waveguides under excitation of CPL[4,5], which offers a versatile toolbox for efficient signal transmission and reception[6], information encoding and retrieval[7], and the construction of optical isolators and circulators for quantum optical applications[8]. However, there are still challenges to be addressed[9]. First, the output intensities at both ports of achiral waveguides are symmetric under linearly polarized excitation. Achieving robust directional transmission through asymmetric nanostructures thus becomes imperative when incident polarization is constrained to a fixed linear state. Second, precise manipulation of the near-field polarization state of light and its interaction with surface plasmon polariton (SPP) modes in plasmonic waveguides remains difficult. Chiral plasmonic nanostructures, which possess geometric properties that cannot be superposed onto their mirror images, have

emerged as a promising solution to address the aforementioned challenges[10]. A seed-mediated growth strategy has been developed to synthesize various chiral Au nanoparticles with controllable sizes and large extinction dissymmetry factors (g-factors) of 0.2[11-13]. These chiral Au nanoparticles exhibit distinct interactions with left- and right-handed circularly polarized (LCP and RCP) light, enabling the modulation of the polarization of the surrounding electromagnetic field. Chiral nanoparticles can be assembled onto the surface of a waveguide and used as spin-filters in routing devices. In addition, precise control over the plasmonic properties of chiral nanoparticles can be achieved by the design of their compositions and sizes. This enables efficient coupling of light to SPPs in the waveguide. Such a chirality-dependent manipulation method plays a crucial role in achieving desired routing performances and meeting specific application requirements. Assembling chiral plasmonic nanoparticles with nano-waveguides is therefore highly desirable, yet this important scenario has remained unexplored.

Chiral plasmonic nanostructures have been integrated with two-dimensional (2D) materials for the manipulation of their luminescent emissions, leveraging the plasmonic and chiroptical properties of

[1]Department of Physics, The Chinese University of Hong Kong, Hong Kong SAR, China. [2]School of Science and Engineering, The Chinese University of Hong Kong, Shenzhen, Guangdong, China. [3]College of Electronics and Information Engineering, Shenzhen University, Shenzhen, Guangdong, China. [4]These authors contributed equally: Yilin Chen, Yang Chen. ✉e-mail: zhuoxiaolu@cuhk.edu.cn; jfwang@phy.cuhk.edu.hk

chiral plasmonic nanostructures and the electronic and optical properties of 2D materials[14]. The recent discovery of 2D materials has provided rich opportunities to explore valley-polarized exciton emissions[15]. The combination of large spin splitting and time-reversal symmetry in transition metal dichalcogenide (TMDC) monolayers leads to opposite signs of spin splitting in the +K and −K valleys. This property enables the selective pumping of electrons into specific valley and spin states using LCP and RCP light in plasmon–exciton hybrid nanostructures[16,17]. Such a valley-dependent optical selection rule results in valley-polarized exciton emissions with particular circular polarization states[18]. Researchers have utilized chiral plasmonic nanostructures to tailor the valley-polarized photoluminescence (PL) in TMDC monolayers at room temperature[19–21]. The foundations are mainly based on two points. First, chiral plasmons can be used to reshape the polarization state of the electromagnetic field at TMDC monolayers. The pumping probability of electrons in one valley can be selectively enhanced or suppressed by the chiral electromagnetic field through the interaction between plasmons and excitons. Second, the Purcell effect induced by chiral plasmons can reduce the recombination lifetime of excitons and effectively increase the quantum yield of an emitter while keeping the valley scattering time unaffected. However, on-chip TMDC-based valleytronic devices face challenges in efficiently routing valley-polarized emissions with large degrees of valley polarization, high directionality, position-independent robustness, and long propagation distances for future applications in information processing and transmission. Coupling a plasmon–exciton hybrid waveguide with a chiral plasmonic nanoparticle is a promising approach to address these challenges. The integrated waveguide can enhance the degree of valley polarization and transduce chirality-dependent valley information into optical spin angular momentum information that can be detected over a long distance. The capabilities provided by the plasmon–exciton hybrid waveguide hold profound implications for the development of valley-based optoelectronics[22], ultrafast valley-polarized transport[23], and valley-based qubits for quantum computing[24].

Herein, we report on the assembly of chiral Au nanocubes (NCs) onto the surface of Ag nanowires (NWs) to achieve the routing of SPPs and PL in WS$_2$ monolayers. The photonic spin-Hall effect is demonstrated with the (chiral Au NC)–NW hybrid waveguides under excitation of CPL. When the system is excited with linearly polarized laser light, the SPPs propagate preferentially in a specific direction along the NW, dictated by the chiroptical response of the attached chiral Au NC. The (chiral Au NC)–NW structures are further used to control the valley-polarized exciton emissions in WS$_2$ monolayers and enhance the degree of circular polarization (DCP) under excitation of linearly polarized light. The pumping and decay rates of excitons located in the elliptically polarized electric field generated by the (chiral Au NC)–NW structures are calculated to investigate the preferential excitation and emission of excitons in one of the valleys. The polarization states of the localized electromagnetic field and the valley states of excitons can be read out by the photonic path of the NW based on the photonic spin-Hall effect. The ability to manipulate the coupling of light and excitons in a chirality-dependent manner using chiral plasmonic nanoparticles provides promising avenues for exploiting spin or valley degrees of freedom to achieve chiral sensors and valleytronic logic gates.

## Results

### Surface plasmon propagation excited by CPL and linearly polarized light

The (chiral Au NC)–NW hybrid waveguides were fabricated by assembling chiral Au NCs with distinct chiral morphologies on as-prepared Ag NWs that were deposited on Si/SiO$_2$ substrates (Supplementary Figs. 1 and 2). D- and L-handed Au NCs were synthesized through seed-mediated overgrowth on Au nano-octahedra in the presence of chiral molecules, which directed asymmetric growth on

high-Miller-index facets. The Ag NW with a single attached chiral Au NC can be found on the Si/SiO$_2$ substrate (Fig. 1a, b). The coupled SPPs were launched as evidenced by the light spot at each end of the NW when a laser beam was focused on the attached chiral Au NC. The left and right outputs are designated by LO and RO, which correspond to the −x direction and +x direction, respectively. The (chiral Au NC)–NW devices with the chiral Au NC located roughly at the center of the Ag NW were fabricated and considered ideal chiral waveguides. These ideal waveguides can eliminate deviations in the LO and RO intensities induced by the different distances from the central position to the LO and RO positions because the output intensity of SPPs decays as the propagating distance along the Ag NW is increased. The intensity difference between the LO and RO in the ideal waveguide can thus be utilized to analyze the coupling efficiency of light propagating along the −x and +x directions. A coordinate system is defined for the fabricated (chiral Au NC)–NW structures. The x-axis is defined to be aligned along the longitudinal direction of the Ag NW. The chiral Au NC is defined to be always positioned on the +y side of the NW.

The chiral Au NC in a (chiral Au NC)–(Ag NW) structure was illuminated by a LCP or RCP laser beam at the wavelength of 633 nm (Fig. 1c). The scattered laser light at the junction of the hybrid structure induced strong light–matter interaction at the surface of the Ag NW, which leads to the excitation of SPP modes within the Ag NW. The LCP excitation led to main SPP emissions at the RO of the (D-handed chiral Au NC)–NW structures, whereas the RCP excitation led to main SPP emissions at the LO, as shown in the pseudocolor images in Fig. 1c (central column). Such a relationship between the propagating direction and the polarization of the incident light is called spin-direction locking or photonic spin-Hall effect. The (L-handed chiral Au NC)–NW structures under the CPL excitation exhibited a similar photonic spin-Hall effect, as evidenced by the pseudocolor images in Fig. 1c (right column). SPPs propagated to one end of the NW preferentially when the CPL was focused on the chiral Au NC.

The scattering g-factors of the individual chiral Au NCs supported on SiO$_2$ substrates characterize the chiroptical responses of the chiral Au NCs[25]. The D- and L-handed chiral Au NCs scatter LCP and RCP light preferentially at the wavelength of 633 nm, respectively. A linearly polarized light can be decomposed into an equal amount of the LCP and RCP components. When the chiral Au NC of the (chiral Au NC)–NW structure is excited by 633 nm linearly polarized laser light, it can scatter light of a specific handedness preferentially. The SPP propagation of the (D/L-handed chiral Au NC)–NW structures under the excitation of 633 nm linearly polarized laser light was demonstrated (Fig. 1d). The incident laser light propagated along the −z direction, with its polarization along the y-axis. The results show that a D-handed chiral Au NC gives a larger SPP intensity at the RO, while the L-handed chiral Au NC gives a larger SPP intensity at the LO. The scanning electron microscopy (SEM) images of the (chiral Au NC)–NW structures with a D-handed chiral Au NC and an L-handed chiral Au NC, respectively, are shown in Supplementary Fig. 3.

The directionality of SPP propagation in the hybrid structure is defined as $(I_R-I_L)/(I_R + I_L)$, where $I_R$ and $I_L$ are the intensities measured at the RO and LO, respectively. The normalized intensities of the LO and RO collected from three representative (D-/L-handed chiral Au NC)–NW devices excited separately by LCP and RCP laser light are summarized in Fig. 1e. The selected regions of the LO and RO in the charge-coupled device (CCD) images and the calculation of the LO and RO intensities are shown in Supplementary Fig. 4. The statistical directionality of the (D-/L-handed chiral Au NC)–NW shows good routing behaviors with a maximal directionality of -1.0 under the excitation of LCP light, whereas a minimal directionality of about −1.0 was obtained under the excitation of RCP light. The spin-direction locking phenomenon under CPL is repeatable. These findings suggest that the chiroptical effect of the attached chiral nanoparticles is overshadowed by the handedness of the incident CPL. The coupling

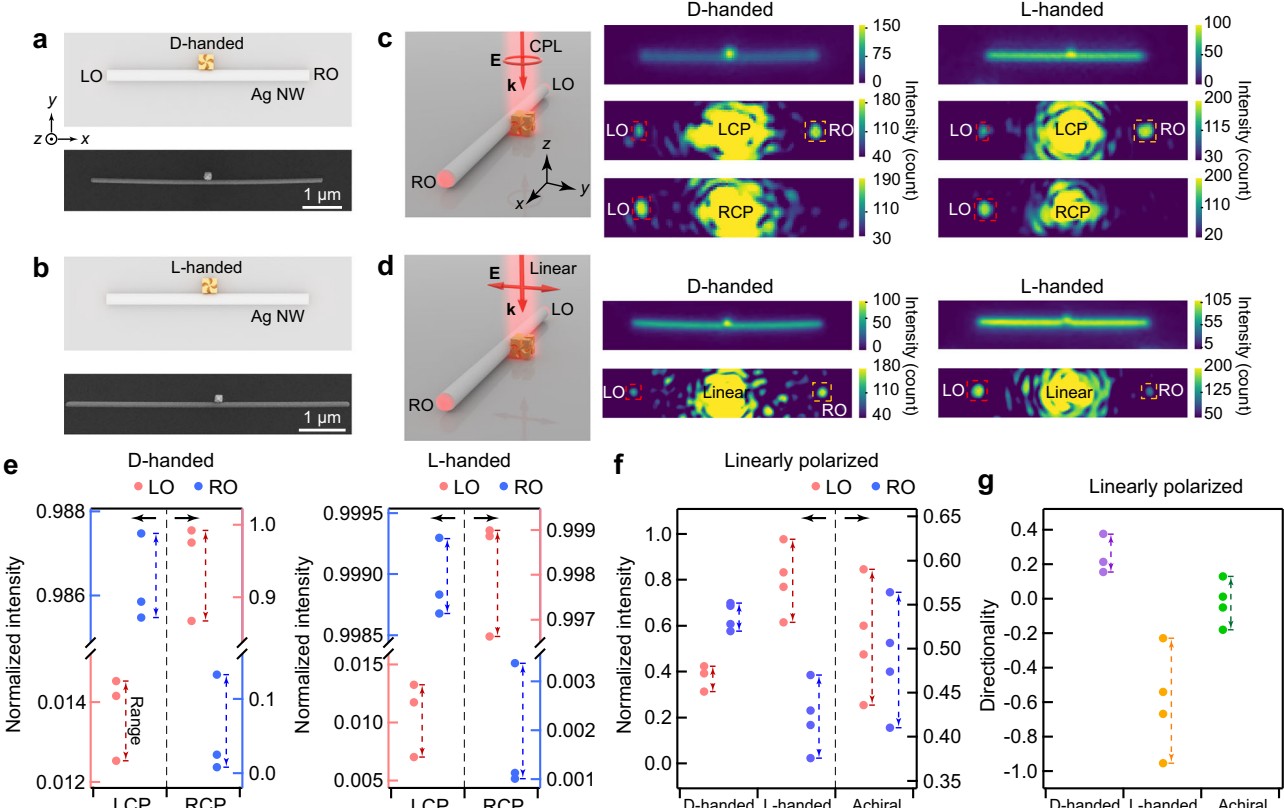

**Fig. 1 | Photonic spin-Hall effect in the (chiral Au nanocube)–(Ag nanowire) structures.** Schematics and scanning electron microscopy images of a D-handed (**a**) and a L-handed nanocube–nanowire (NC–NW) structure (**b**). **c** Schematic and pseudocolor images of the (chiral Au NC)–NW structures under the excitation of 633 nm circularly polarized light (CPL). The incident laser light propagated along the −z direction. The left- and right-handed circularly polarized laser lights were coupled toward the right output (RO, yellow box) and left output (LO, red box) of the NW, respectively. The pseudocolor dark-field scattering images of the structures are positioned above the corresponding laser-on images. **d** Schematic and pseudocolor images of the (chiral Au NC)–NW structures under the excitation of

633 nm linearly polarized light. **e** Normalized LO and RO intensities of the (D-/L-handed chiral Au NC)–NW structures under the excitation of CPL. The data were collected from 3 (D-/L-handed chiral Au NC)–NW structures, respectively. Normalized intensities of the LO and RO (**f**) and directionality of the propagation (**g**) in the (D-/L-handed chiral Au NC)–NW structures and (achiral Au NC)–NW structures under the excitation of linearly polarized light. The data were collected from 4 (D-/L-handed chiral Au NC)–NW structures and 4 (achiral Au NC)–NW structures, respectively. The dashed arrows in (**e**–**g**) represent the range of measured normalized intensities.

between the chiral Au NC and the Ag NW enhances the electric field at their junction while maintaining the polarization of the incident light. The LCP and RCP components of light in the gap region between the chiral Au NC and the Ag NW are coupled into the SPP modes with different propagation directions according to the photonic spin-Hall effect. The photonic spin-Hall effect was not violated when the chiral Au NC and the laser spot were moved away from the central position of the Ag NW and the attached chiral Au NCs were subjected to spatial rotations relative to the ideal chiral waveguide (Supplementary Fig. 5). More (chiral Au NC)–(Ag NW) structures were further fabricated and excited with linearly polarized laser light to evaluate the routing performance (Fig. 1f, g). The output intensities and directionality of the structures were calculated in the way described above. The chirality-dependent directional propagation of light can be reproduced. The RO intensities are higher than the LO intensities in all D-handed devices, while the LO intensities exceed the RO ones in all L-handed devices. The statistical results show that the directionality of SPP propagation fluctuates with the morphologies of the attached chiral Au NCs. A minimal directionality of about −0.96 for the L-handed devices and a maximal directionality of -0.4 for the D-handed devices were obtained under the excitation of linearly polarized light.

The L-handed chiral Au NC of the hybrid structure was then excited by 633 nm linearly polarized laser light with polarization aligned parallel to the long axis of the Ag NW (Supplementary Fig. 6a).

The intensity of the LO was higher than that of the RO. Rotating the polarization direction of the linearly polarized laser light by 90° still permitted the observation of the chirality-dependent directional propagation in the (chiral Au NC)–NW structure. Waveguides with chiral elements can be oriented in different directions relative to the linear polarization direction of incident light to achieve routing, whereas waveguides with achiral elements require a specific orientation or position relative to the direction of incident CPL for routing. In addition, only the light spot from the LO could be detected when the attached L-handed chiral Au NC was spatially rotated with respect to the ideal devices (Supplementary Fig. 6b, c), which indicates the maintenance of the chirality-dependent directional propagation in the structure. To further make a comparison with the results from the chiral structures, the SPP propagation in achiral structures was investigated (Fig. 1f, g and Supplementary Figs. 7–9). An achiral Au NC was assembled onto the surface of the Ag NW. LCP and RCP exciting light were directionally coupled to the LO and RO of the NW, respectively, which is referred to as a reversed photonic spin-Hall effect. Such a reversal highlights the role of plasmon coupling in spin-dependent routing. Under the excitation of linearly polarized light, the intensity difference between the LO and RO was negligible. The directional SPP propagation observed in the (chiral Au NC)–NW structures under linearly polarized light, therefore, originates from their chiroptical responses and the photonic spin-Hall effect.

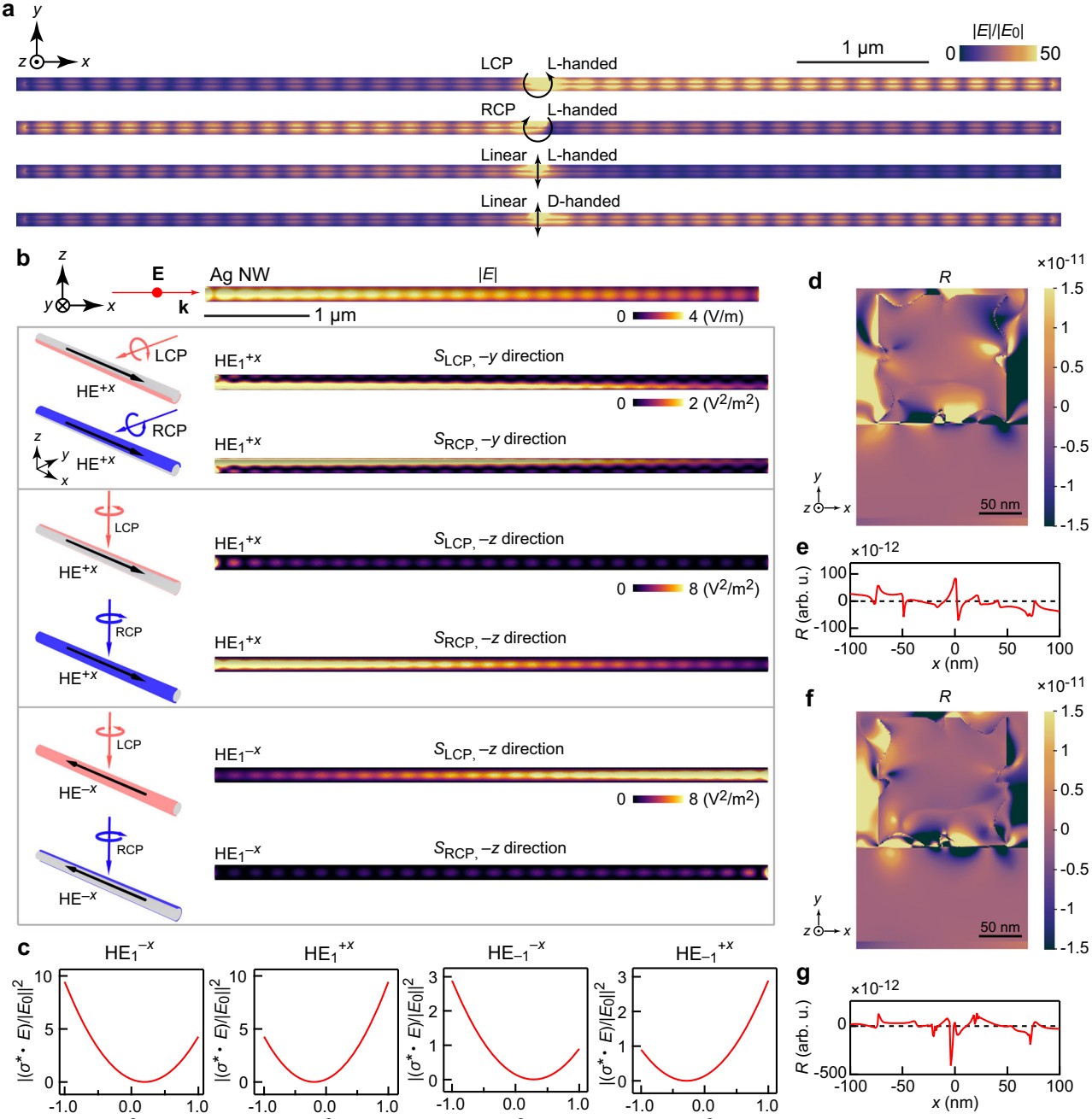

**Fig. 2 | Theoretical demonstration of the photonic spin-Hall effect.**
**a** Distributions of the electric field at the bottom $x$–$y$ plane surface of the L-/D-handed chiral Au nanocube–nanowire (NC–NW) structures under the excitation of circularly and linearly polarized dipoles, respectively. **b** Distributions of the electric field ($|E|$) and the overlaps between the $HE_1$ mode and circularly polarized light. The $HE_1^{+x}$ and $HE_1^{-x}$ modes correspond to the propagating mode along the $+x$ and $-x$ directions, respectively. **c** Variation of overlapping intensities between the elliptically polarized light illumination along the $-z$ direction and the HE modes ($S_\sigma/|E_0|^2$) with polarization ellipticity $\varepsilon$. Distribution of the strength of the ellipticity ($R$) in the $x$–$y$ plane region of interest (ROI) (**d**) and variation of $R$ with respect to the $x$-coordinate in the line drawn at the gap region which is 1 nm away from the surface of the NW (**e**) of the (L-handed chiral Au NC)–(Ag NW) structure. Distribution of $R$ in the $x$–$y$ plane ROI (**f**) and variation of $R$ with respect to the $x$-coordinate in the line (**g**) of the (D-handed chiral Au NC)–(Ag NW) structure.

## Theoretical demonstration of the photonic spin-Hall effect

The distributions of the electric field in the NW of the (chiral Au NC)–(Ag NW) structure under the excitation of 633 nm circularly polarized and linearly polarized dipoles were simulated by the finite-difference time-domain (FDTD) method (Fig. 2a). The propagating SPPs excited toward the $+x$ direction appear to be stronger than those toward the $-x$ direction under the illumination of LCP light. On the other hand, SPPs are more likely to propagate toward the $-x$ direction under illumination of RCP light. The simulations for the (D-handed

chiral Au NC)–(Ag NW) structure gave similar results (Supplementary Fig. 10), indicating that the photonic spin-Hall effect does not change with the chirality of the attached chiral Au NC. In contrast, the propagating direction of SPPs excited by linearly polarized dipoles is reversed when the attached chiral nanoparticle changes from L-handed to D-handed (Fig. 2a). The propagating SPPs preferentially propagate to the $-x$ direction and the $+x$ direction for an L-handed and D-handed chiral Au NC, respectively. The cross-sections of the Ag NWs were experimentally characterized as a smoothed pentagon

(Supplementary Fig. 11a). The FDTD simulations of the (chiral Au NC)–(faceted pentagonal Ag NW) structures suggest that the spin-direction locking effect and the directional SPP propagations induced by the chirality of the attached chiral nanoparticle remain robust (Supplementary Fig. 11b–d). The simulations of the achiral hybrid structures demonstrated that the direction of SPP propagation under the excitation of CPL can be tuned by the plasmon coupling of the NW with different attached nanoparticles (Supplementary Fig. 12).

The photonic spin-Hall effect was theoretically demonstrated by calculating the intensities scattered into the plasmon modes of the Ag NW. The overlapping intensities for the incident LCP and RCP light can be determined by $S_{LCP} = |\sigma_{LCP}^* \cdot \mathbf{E}|^2$ and $S_{RCP} = |\sigma_{RCP}^* \cdot \mathbf{E}|^2$, respectively[5], with $\sigma_{LCP}$ and $\sigma_{RCP}$ representing the polarization states of CPL. The theoretical calculation process of the fundamental modes[26,27] and overlapping intensities of an infinitely long Ag NW are shown in Supplementary Figs. 13 and 14 and Supplementary Note 1. The propagating plasmon modes of an Ag NW with a specific length and the overlapping intensities were reproduced by COMSOL and FDTD simulations. The hybrid modes, $HE_1$ and $HE_{-1}$ mode, of the Ag NW can be excited by linearly polarized light with $y$-direction and $z$-direction polarizations (Supplementary Fig. 15). The superscripts $+x$ and $-x$ of the HE modes represent the propagation direction of the SPP modes. The calculated overlaps localized on the surface of the NW show that the photonic spin-Hall effect is valid for the coupling between CPL illumination along the short axis of the NW and the HE modes (Fig. 2b and Supplementary Figs. 16 and 17). When the incident electromagnetic field with LCP and $-z$ (or $-y$) propagating direction is localized at the $+y$ (or $-z$) side of the NW surface, the incident field will be efficiently coupled into the $HE_1^{+x}$ mode, and thus the SPPs will propagate along the $+x$ direction. The $+x$ direction means that the SPPs will be routed to the RO of the Ag NW. When the incident electromagnetic field with RCP and $-z$ propagation direction is localized at the $+y$ side of the NW surface, the incident field efficiently couples into the $HE_1^{-x}$ mode and the excited SPPs propagate along the $-x$ direction, corresponding to the LO of the NW. Such a mirror symmetry of $S_{LCP}$ and $S_{RCP}$ is also valid for the distributions of the overlapping intensities for CPL propagating along the $-y$ or $-z$ directions and the $HE_{-1}$ mode (Supplementary Fig. 17). The mirror symmetry of $S_{LCP}$ and $S_{RCP}$ for the HE modes was reproduced by FDTD simulations and the calculations on the overlapping intensities (Supplementary Fig. 18 and Supplementary Note 2). The coupling efficiency of the plasmon modes in the NW and CPL is therefore determined by the polarization and the distribution of the localized electromagnetic field enhancement on the surface of the NW. The coupling of the nanoparticle and the NW modifies the directions of the Poynting vectors in the gap between the nanoparticle and the NW. Further analysis of the $z$-component of the Poynting vector ($P_z$) in Supplementary Fig. 19 reveals that the sign reversal of $P_z$ from negative to positive accounts for the transition from the normal to the reversed spin-Hall effect. The theoretically demonstrated photonic spin-Hall effect can be used to explain the directionality of the SPP outputs when a chiral Au NC is positioned on the $+y$ side of the NW surface and illuminated with a focused laser beam.

We also analyzed the photonic spin-Hall effect of the Ag NW coupling with elliptically polarized light by calculating the overlapping intensities. An elliptically polarized light propagating along the $-z$ direction can be written as $\sigma = (-\mathbf{e}_x - i\,\varepsilon \mathbf{e}_y)/\sqrt{2}$. The polarization ellipticity $\varepsilon$ can take values between $-1$ (RCP) and $+1$ (LCP). The polarization of light becomes linear when $\varepsilon$ is equal to 0. $S_\sigma$ is defined as $|\sigma^* \cdot \mathbf{E}(y,z)|^2$. The $S_\sigma/|E_0|^2$ values between linearly polarized light and the $HE_1$ mode are distributed separately and accumulated on the two sides of the NW, resulting in the malfunction of the photonic spin-Hall effect under the excitation of linearly polarized light (Supplementary Fig. 20). The overlaps between the $HE_1^{-x}$ mode and elliptically polarized light on the $+y$ side of the NW surface decreases as $\varepsilon$ is increased from $-1$ (RCP) to 0 (Fig. 2c). The overlaps between the $HE_1^{+x}$ mode and elliptically

polarized light on the same side of the NW decreases as $\varepsilon$ is decreased from $+1$ (LCP) to 0. The variation of $S_\sigma/|E_0|^2$ on the $+y$ side of the NW surface for the $HE_{-1}^{-x}$ and $HE_{-1}^{+x}$ mode, respectively, as a function of $\varepsilon$ show similar results (Fig. 2c). The photonic spin-Hall effect under the excitation of elliptically polarized light only occurs when its ellipticity has a relatively large absolute value. The preferred propagation direction of SPPs in the hybrid structure can therefore be predicted by analyzing the distribution of the ellipticity at the junction and the photonic spin-Hall effect. The distributions of the ellipticity of the (chiral Au NC)–(Ag NW) structures were calculated to analyze the chirality-dependent SPP propagation. A 633 nm linearly polarized Gaussian beam with polarization perpendicular to the long axis of the Ag NW was illuminated on the chiral Au NC (Fig. 2d–g and Supplementary Figs. 21 and 22). The electric field is enhanced in the gap region between the chiral Au NC and the Ag NW. The coupling effect results in a strongly scattered electromagnetic field that can be efficiently coupled into the SPP modes of the Ag NW. The correlation between the ellipticity of light in the gap region, with the wavevector pointing toward the $-z$ direction, and the routing effect was understood based on the calculation results. The distribution of the $z$-component of the Poynting vector $P_z$ in the $x$–$y$ plane ROI is shown in Supplementary Fig. 21c. The ellipticity angle $\eta$ of the electric field in the $x$–$y$ plane can be defined as

$$\tan \eta = \tan\left(\frac{\arcsin\left(\frac{S_3}{S_0}\right)}{2}\right) = \frac{E_y}{E_x} \qquad (1)$$

by use of the Stokes parameters of $S_0 = |E_x|^2 + |E_y|^2$ and $S_3 = 2\mathrm{Im}(E_x^* E_y)$. The distribution of $\tan\eta$ shows that the electromagnetic field in the gap region between the chiral Au NC and the Ag NW is elliptically polarized (Supplementary Fig. 21d). The strength of the ellipticity $R$ in the gap region defined as $-(\tan \eta)\,P_z$ was used to analyze the routing effect of the (chiral Au NC)–(Ag NW) structures (Fig. 2d). The negative sign defines the direction of $P_z$ to be pointing in the $-z$ direction, which corresponds to the previously defined incident light direction. The sum of $R$ in the line has a negative value of $-5.8 \times 10^{-10}$ for the (L-handed chiral Au NC)–NW structure, which indicates that the RCP component is dominant (Fig. 2e). The resultant right-handed elliptically polarized electromagnetic field in the gap region leads to a higher LO intensity, in accordance with the observed chirality-dependent directional propagation.

The distributions of $P_z$, $\tan\eta$, and $R$ in the $x$–$y$ plane monitor were also calculated for the (D-handed chiral Au NC)–(Ag NW) structure using the same method as for the L-handed one (Fig. 2f and Supplementary Fig. 22). The sum of $R$ in the line has a positive value of $7.8 \times 10^{-10}$ for the (D-handed chiral Au NC)–NW structure, which indicates that the LCP component is dominant (Fig. 2g). The resultant left-handed elliptically polarized electromagnetic field in the gap region leads to a higher RO intensity. The handedness of the elliptically polarized electromagnetic field in the gap region induced by the D-handed chiral Au NC and L-handed chiral Au NC is opposite. The opposite handednesses originate from their different twisting directions of the chiral arms and concave surfaces. The SPPs in the NW can therefore be routed under the excitation of linearly polarized light depending on the chiroptical properties of the attached chiral nanoparticle. To investigate the effect of the distance between the chiral Au NC and the Ag NW on the SPP propagation, the chiral Au NCs were coated with a 15-nm-thick $SiO_2$ shell (Supplementary Fig. 23). The $SiO_2$ shell modulated the plasmon coupling and thus reversed the directionality compared to the results of the (chiral Au NC)–NW structures. The simulation results reveal that tuning the $SiO_2$ thickness from 10 nm to 5 nm can switch the SPP propagation direction under linearly polarized excitation (Supplementary Fig. 24). The SPP routing depends not only on the chiroptical property of the nanoparticle but also on its plasmon coupling with the NW.

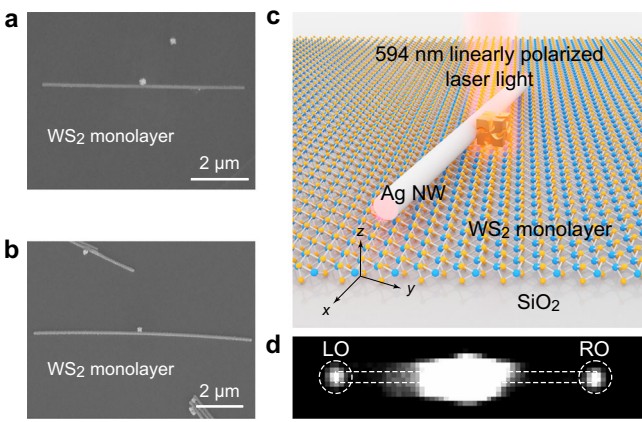

**Fig. 3 | Fabrication and surface plasmon polariton excitation of the (chiral Au nanocube)–(Ag nanowire) structures on WS₂ monolayers.** Scanning electron microscopy images of a D-handed (**a**) and a L-handed nanocube–nanowire (NC–NW) structure on WS₂ monolayers (**b**). **c** Schematic of the (chiral Au NW)–(Ag NW)-on-WS₂ structure under the excitation of linearly polarized laser light. **d** Photoluminescence (PL) image for the laser illumination on the chiral Au NC. The white dashed circles indicate the PL output signals. The white dashed rectangle represents the profile of the NW.

## Chirality-dependent routing of the valley excitons in WS₂ monolayers

The coupling of plasmons and valley excitons was investigated using the (chiral Au NC)–(Ag NW) structures combined with WS₂ monolayers. The WS₂ monolayers were grown by physical vapor deposition[28]. The (chiral Au NC)–(Ag NW)-on-WS₂ structures were fabricated through a transfer process (Fig. 3a, b and Supplementary Fig. 25). The details of the fabrication process of the (chiral Au NW)–(Ag NW)-on-WS₂ structures are provided in Methods. The (chiral Au NC)–(Ag NW) structures were utilized to alter the excitation and emission of nearby excitons in the WS₂ monolayer (Fig. 3c). The localized surface plasmon resonance effect induced by the chiral Au NC resulted in an increase in the PL intensity of the WS₂ monolayer when a laser beam was focused onto the chiral Au NC. The valley-polarized excitons in the WS₂ monolayer can be considered as circular transition dipoles to couple with the SPP modes of the Ag NW[29,30]. The SPPs will propagate along the Ag NW and emit light at the ends. The experimental setup for the measurements of the PL spectra, as well as the LCP and RCP components of the PL emissions, is illustrated in Supplementary Fig. 26a. The PL spectra at the LO and RO of the Ag NW were measured simultaneously under the excitation of linearly polarized laser light propagating along the −z direction (Fig. 3d).

The correlation between the directionality of the SPP emissions and the chiroptical response of the chiral nanoparticle was investigated by examining the routing performance under the excitation of linearly polarized laser light at the wavelengths of 594 nm and 514 nm. The polarization direction of the excitation light for the PL emission measurements was aligned along the x-axis. The PL spectra at the outputs of the (chiral Au NC)–(Ag NW)-on-WS₂ structure under the excitation of linearly polarized laser light were measured (Fig. 4a–d). The PL intensity of the RO at the peak wavelength ($PL_{RO}$) is stronger than that of the LO ($PL_{LO}$) under the excitation of 594 nm light (Fig. 4a) for the D-handed structure. The PL spectra of the L-handed structure under the excitation of 514 nm light show that $PL_{LO}$ is higher than $PL_{RO}$ (Fig. 4d). The directionality of the SPP emissions can be expressed by $(PL_{RO} - PL_{LO}) / (PL_{RO} + PL_{LO})$. The PL emission directionality is positive for the D-handed structure, whereas the routing direction is reversed with the L-handed structure (Fig. 4e). The PL spectra were also measured from NW-on-WS₂ structures (Supplementary Fig. 26b). The dashed lines in Fig. 4a show that $PL_{LO}$ and $PL_{RO}$ for the NW-on-WS₂

structure have nearly the same intensity, resulting in a near-zero directionality.

The scattering spectra of the attached chiral Au NCs in the (chiral Au NC)–(Ag NW)-on-WS₂ structures were measured under illumination of CPL (Supplementary Fig. 27). The corresponding scattering g-factor spectra were utilized to explore the chiroptical responses of the chiral Au NCs at different excitation wavelengths. The scattering intensity of the attached D-handed chiral Au NC under LCP light is greater than that under RCP light, resulting in a positive scattering g-factor. In contrast, the scattering g-factor of the attached L-handed chiral Au NC is negative. The sign and magnitude of the scattering g-factor reflect the handedness and strength of the chiroptical response, which governs the spin-selective coupling of linearly polarized incident light into SPPs. A larger positive g-factor at a given wavelength indicates stronger preferential scattering of LCP light, which leads to more pronounced directional propagation of SPPs toward the RO of the NW. A negative scattering g-factor implies dominant RCP scattering and directional outputs toward the LO. The directionalities under the excitation of 594 nm and 514 nm laser light in Fig. 4e ($D_{594}$ and $D_{514}$, in short) and the scattering g-factors at the excitation wavelengths of 594 nm and 514 nm ($g_{594}$ and $g_{514}$, in short) are represented in Fig. 4f. The variations in $D_{594}$ and $D_{514}$ are correlated with the $g_{594}$ and $g_{514}$ values of the attached chiral Au NCs, respectively. The correlation stems from the chiral plasmonic resonance. The scattering g-factor peak at the wavelength of 593 nm indicates strong chiroptical on-resonance with 594 nm excitation. The 594 nm linear polarization excitation therefore efficiently enables a spin-selective coupling into directional SPPs, yielding a high $D_{594}$. In contrast, the much smaller g-factor at 514 nm ($g_{514}$) reflects off-resonant chiroptical interaction. This results in a less efficient linear-to-elliptical polarization conversion and a trivial directionality ($D_{514}$). The higher $|g_{514}|$ value than the $|g_{594}|$ value for the L-handed chiral Au NC results in higher $|D_{514}|$ than $|D_{594}|$ (Fig. 4f). The variation of the directionality of the SPP emissions is therefore dependent on the chiroptical response of the attached chiral Au NC.

The LCP- and RCP-resolved PL spectra and DCPs of the PL outputs were further investigated under the excitation of 594 nm and 514 nm linearly polarized laser light (Fig. 5 and Supplementary Fig. 28). The DCP of the PL can be expressed by $(PL_{LCP} - PL_{RCP})/(PL_{LCP} + PL_{RCP})$, with $PL_{LCP}$ and $PL_{RCP}$ representing the LCP and RCP components of the PL at the peak wavelength, respectively. For the (Ag NW)-on-WS₂ structure, $PL_{RCP}$ at the LO is stronger than $PL_{LCP}$ (Fig. 5a). $PL_{LCP}$ at the RO is stronger than $PL_{RCP}$. The DCP of the PL at the LO is negative, while the DCP of the PL at the RO is positive (Fig. 5b). The DCPs of the LO for both L-handed and D-handed structures are positive, while the DCPs of the RO are negative (Fig. 5d, f). The absolute DCP values of the ROs are higher than those of the LOs for both the L-handed and D-handed structures. To further clarify the essential role of chiral plasmonic nanoparticles, we conducted a series of control experiments using a bare (Ag NW)-on-WS₂ structure without any attached nanoparticle and (achiral Au NC)–NW-on-WS₂ structures (Supplementary Figs. 29–31). In all control cases, whether no particle, a single achiral Au NC, or an achiral Au NC dimer was attached, the PL intensities from the LO and RO showed negligible differences under linear polarization excitation, and the DCP remained close to zero. These results confirm that the chiral Au nanoparticle is essential for achieving the observed directional routing of PL. In contrast, achiral hybrids or the NW alone do not support directional output under the same excitation geometry.

## Understanding of the chirality-dependent routing of valley-polarized emissions

Once valley excitons in the hybrid structures are pumped and excited, they can recombine through direct radiation, SPP excitation, and nonradiative damping, including intrinsic nonradiative decay and metal-induced nonradiative decay. The modulated DCP of valley-polarized emissions depends on the modulated pumping rate and the

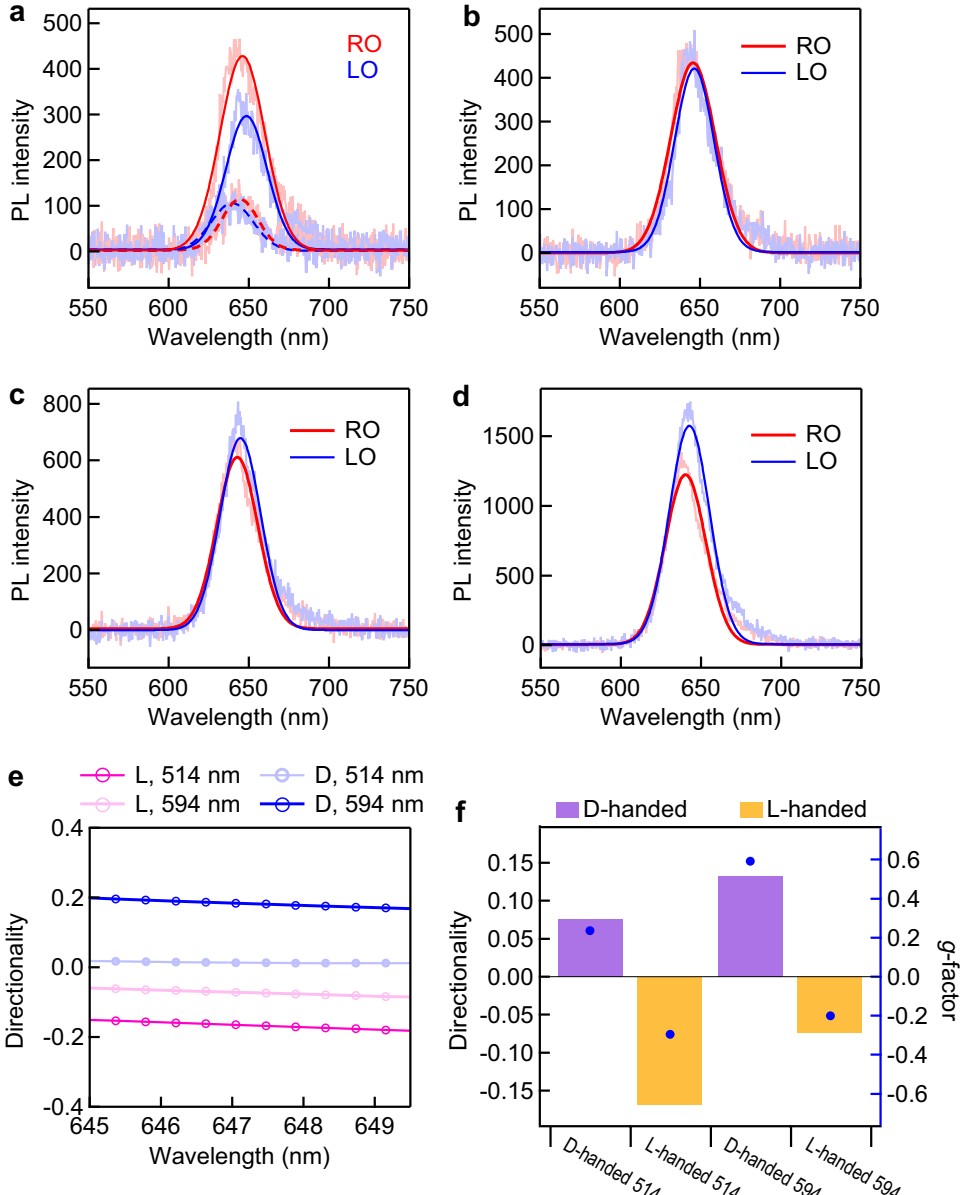

**Fig. 4 | Valley emissions with the (chiral Au nanocube)–(Ag nanowire) structures on WS$_2$ monolayers under the excitation of linearly polarized laser light.** Photoluminescence (PL) spectra of the left and right outputs (LO and RO) with the D-handed structure under the excitation of 594 nm (**a**) or 514 nm (**b**) laser light. The dashed lines in (**a**) represent the PL spectra of the LO and RO of the Ag nanowire (NW) on a WS$_2$ monolayer. PL spectra of the output with the L-handed structure under the excitation of 594 nm (**c**) or 514 nm (**d**) laser light. The smooth lines in (**a–d**) are the Gaussian fits of the experimental data. **e** Directionalities of the surface plasmon polariton (SPP) emissions in the (chiral Au nanocube)–(Ag NW)-on-WS$_2$ structures. **f** Directionalities of the SPP emissions (bar graph) and scattering $g$-factors (blue symbols) under the excitation of 514 nm and 594 nm light.

decay rate of excitons in the $+K$ and $-K$ valleys. The modulated pumping rate $P_{LCP}$ of excitons in the $+K$ valley and $P_{RCP}$ of excitons in the $-K$ valley are related to the modulus of the LCP and RCP components of the electric field ($|E_{LCP}|^2$ and $|E_{RCP}|^2$), respectively[18]. $E_{LCP}$ and $E_{RCP}$ can be calculated using

$$E_{LCP} = \frac{1}{2}(E_x - iE_y) \tag{2}$$

$$E_{RCP} = \frac{1}{2}(E_x + iE_y) \tag{3}$$

The distributions of the electric field on the WS$_2$ monolayer were calculated by FDTD. Gaussian beams with the different wavelengths of

594 and 514 nm were, respectively, illuminated at the center of the chiral Au NC along the $-z$ direction. An $x$–$y$ plane monitor, spanning an area of 700 nm × 80 nm, was placed on the surface of the WS$_2$ monolayer. The center of this monitor was aligned with the center of the NW in the $z$ direction. The monitor was equally divided into two ROIs, ROI 1 and ROI 2 (Fig. 6a). The excited SPPs by LCP excitonic emissions in ROI 1 will propagate to the RO, while those excited by RCP excitonic emissions in ROI 1 will route to the LO. The propagation directions of LCP and RCP excitonic emissions in ROI 2 interchange with each other based on the photonic spin-Hall effect. Figure 6a–g and Supplementary Fig. 32 show the simulated results for the (Ag NW)-on-WS$_2$, (D-handed chiral Au NC)–(Ag NW)-on-WS$_2$, and (L-handed chiral Au NC)–(Ag NW)-on-WS$_2$ structures under the excitation of the 594 nm Gaussian beam. For an (Ag NW)-on-WS$_2$ structure without any chiral Au

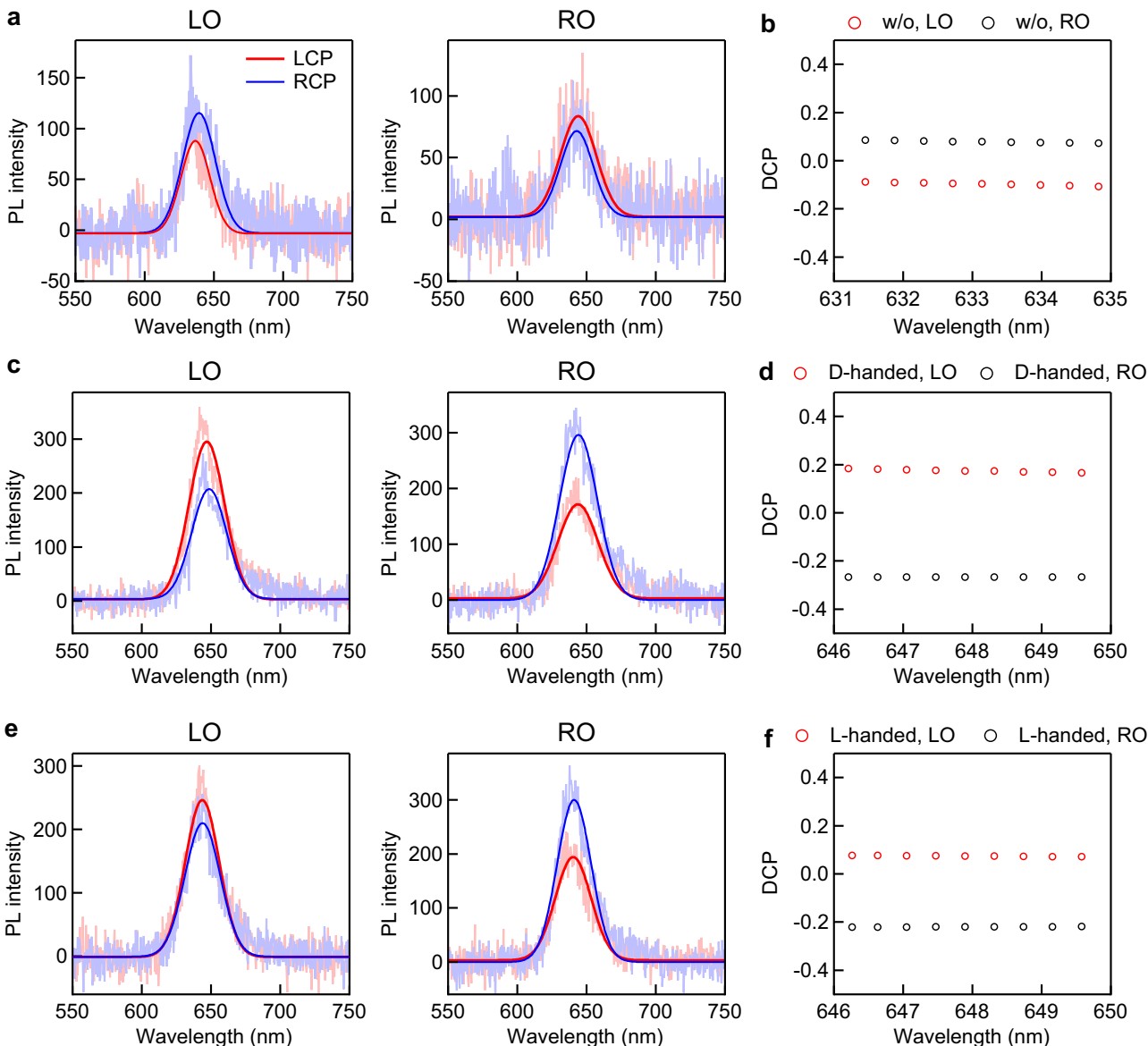

**Fig. 5 | Polarization-resolved photoluminescence spectra and degrees of circular polarization of the photoluminescence signals. a** Polarization-resolved spectra of the photoluminescence (PL) at the right and left outputs (RO and LO) of the nanowire (NW) on a WS$_2$ monolayer under the excitation of 594 nm linearly polarized laser light. **b** Degrees of circular polarization (DCPs) of the PL signals of the (Ag NW)-on-WS$_2$ structure at the peak wavelength. Polarization-resolved spectra and DCPs of the (D-handed chiral Au nanocube)–(Ag NW)-on-WS$_2$ (**c, d**) and (L-handed chiral Au nanocube)–(Ag NW)-on-WS$_2$ structure (**e, f**) under the excitation of 594 nm linearly polarized laser light.

NC, $P_{LCP}$ in ROI 1 is comparable to $P_{RCP}$ in ROI 1 (Supplementary Fig. 32b). In contrast, $P_{RCP}$ in ROI 1 is larger than $P_{LCP}$ in ROI 1 for the L-handed structure (Fig. 6b), while $P_{LCP}$ in ROI 1 is larger than $P_{RCP}$ in ROI 1 for the D-handed structure (Supplementary Fig. 32d). The modulated decay rates of the left-handed and right-handed circular dipoles ($\Gamma_{LCP}$ and $\Gamma_{RCP}$) were simulated, respectively. The circular dipoles were placed below the (chiral Au NC)–(Ag NW) structure in the ROIs. $\Gamma_{LCP}$ and $\Gamma_{RCP}$ were normalized by the decay rate in the absence of any structure to estimate metal-induced nonradiative damping. The modulated pumping rates and decay rates were then multiplied. The obtained $P_{LCP}\Gamma_{LCP}$ and $P_{RCP}\Gamma_{RCP}$ were used to estimate LCP and RCP excitonic emissions, respectively (Fig. 6b and Supplementary Fig. 32b, d).

To compare the $P\Gamma$ differences between the D-handed and L-handed structures more clearly, the distribution of $P\Gamma$ along the x-axis within the ROIs 1 and 2 is presented in Fig. 6c–g. The results show that $P_{RCP}\Gamma_{RCP}$ in ROI 1 is larger than $P_{LCP}\Gamma_{LCP}$ in ROI 1 for the L-handed

structure (Fig. 6c), while $P_{LCP}\Gamma_{LCP}$ in ROI 2 is larger than $P_{RCP}\Gamma_{RCP}$ in ROI 2 (Fig. 6d). The situation is reversed for the D-handed structure (Fig. 6e, f). The relative strength of each type of emission in the different ROIs and the contributions to the overall SPP emissions determine the emission intensities of the LO and RO based on the photonic spin-Hall effect. To discuss the contributions of the $P\Gamma$ in the different ROIs to the overall output of the NW, $(P_{LCP}\Gamma_{LCP})_j$ represents the summation of the $P_{LCP}\Gamma_{LCP}$ over the region $j$, with $j = 1$ or 2, corresponding to ROI 1 or 2. The results of $(P_{LCP}\Gamma_{LCP})_j$ and $(P_{RCP}\Gamma_{RCP})_j$ are presented in Fig. 6g. For both the D-handed and L-handed structures, $(P_{RCP}\Gamma_{RCP})_2$ is larger than $(P_{LCP}\Gamma_{LCP})_1$ for RO, while $(P_{LCP}\Gamma_{LCP})_2$ is larger than $(P_{RCP}\Gamma_{RCP})_1$ for LO. The obtained difference in the LCP and RCP components that contribute to the LO and RO is consistent with the measured results in Fig. 5c–f, showing a positive DCP for the LO and a negative DCP for the RO. The summation of $(P_{LCP}\Gamma_{LCP})_1$ and $(P_{RCP}\Gamma_{RCP})_2$ gives the contributions of the emissions coupled to RO and is designated as $P\Gamma_{RO}$. The summation of $(P_{RCP}\Gamma_{RCP})_1$ and $(P_{LCP}\Gamma_{LCP})_2$

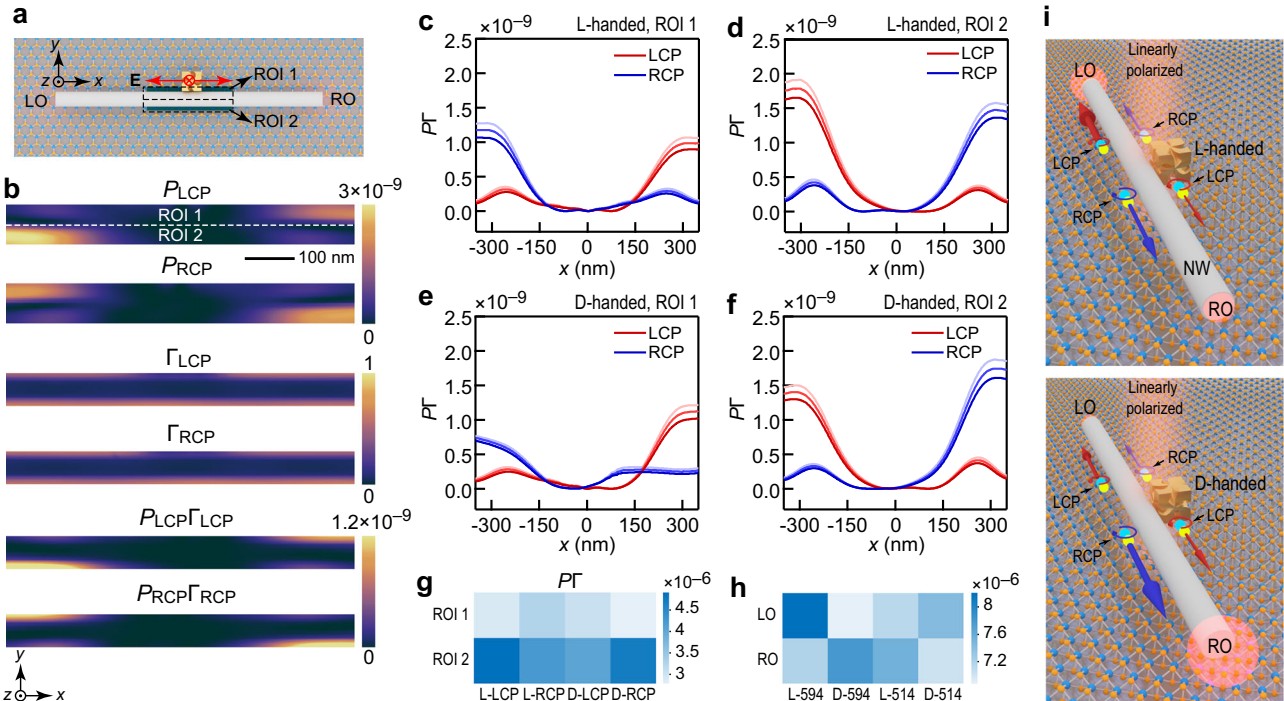

**Fig. 6 | Calculation of the emission efficiencies of valley excitons under the excitation of a linearly polarized Gaussian beam. a** Schematic of a L-handed chiral Au nanocube–nanowire (NC–NW) structures on WS$_2$ monolayers. **b** Distributions of the modulated pumping rates $P_{LCP}$, $P_{RCP}$, $\Gamma_{LCP}$, $\Gamma_{RCP}$, $P_{LCP}\Gamma_{LCP}$, and $P_{RCP}\Gamma_{RCP}$ of the L-handed structure under the excitation of the 594 nm Gaussian beam. Subscript LCP/RCP represents left-/right-handed circularly polarized. Distributions of $P_{LCP}\Gamma_{LCP}$ (red lines) and $P_{RCP}\Gamma_{RCP}$ (blue lines) of the L-handed structure along the three $x$-lines of region of interest (ROI) 1 (**c**) and ROI 2 (**d**). The three $x$-lines in ROI 1 or ROI 2 are located at 31 nm, 29 nm, and 27 nm from the central white dashed line of the entire ROI, respectively, and are represented by different colors from bright to dark. Distributions of the $P_{LCP}\Gamma_{LCP}$ (red lines) and $P_{RCP}\Gamma_{RCP}$ (blue lines) of the D-handed structure along the three $x$-lines of ROI 1 (**e**) and ROI 2 (**f**). **g** Calculated LCP and RCP components of $P\Gamma$ in the ROIs 1 and 2 of the structures. **h** Calculated $P\Gamma$ for the left and right outputs (LO and RO) under the excitation of the 594 nm and 514 nm Gaussian beam. **i** Schematics of the routing of valley excitons by the L- and D-handed structures.

contributes to LO and is designated as $P\Gamma_{LO}$. $P\Gamma_{LO}$ is larger than $P\Gamma_{RO}$ for the L-handed structure under the excitation of the 594 nm Gaussian beam, while $P\Gamma_{RO}$ is larger than $P\Gamma_{LO}$ for the D-handed structure (Fig. 6h, i). The calculated directionality is consistent with the measured results of the routing effect of the PL emissions in Fig. 4a–d. In contrast, $P\Gamma_{RO}$ is comparable to $P\Gamma_{LO}$ for the (Ag NW)-on-WS$_2$ structure (Supplementary Fig. 32b). To compare the directionalities of the structures under different excitation wavelengths, the $P\Gamma_{RO}$ and $P\Gamma_{LO}$ of the D-handed and L-handed structures under the excitation of the 514 nm Gaussian beam are also shown in Fig. 6h. The diminishing difference between $P\Gamma_{LO}$ and $P\Gamma_{RO}$ under 514 nm excitation, compared to the results obtained with 594 nm excitation, is attributed to the weaker chiroptical response. The routing effect of the PL emissions in the (chiral Au NC)–(Ag NW)-on-WS$_2$ structures under the excitation of linearly polarized light is, therefore, originated from chirality-dependent coupling between valley excitons and the modulated chiral electromagnetic field.

As an extended demonstration of chirality-dependent routing, we embedded fluorophore molecules R640 in the SiO$_2$ shell surrounding the chiral Au NC and studied the PL SPP routing of R640 molecules (Supplementary Figs. 33 and 34). Under linearly polarized excitation, the R640 emissions exhibit directional SPP propagation that reverses with the handedness of the attached chiral NC. The D-handed structures route the fluorescence preferentially toward the LO, whereas the L-handed structures route it toward the RO. Under LCP/RCP excitation, the D-handed structures exhibit a more pronounced difference between the LO and RO under LCP than under RCP illumination, while the L-handed structures show a greater output asymmetry under RCP than under LCP illumination. These results confirm that the chiral plasmon–emitter coupling imposes a spin-selective directionality not only on excitonic emissions but also on molecular fluorescence emissions, underscoring the generality of chirality-dependent routing.

## Discussion

The directional propagation of the SPPs in the (chiral Au NC)–(Ag NW) structures has been demonstrated under the excitation of linearly polarized laser light. The maximal absolute value of directionality reaches ~96%, which indicates a great routing performance induced by the strong chiroptical response of the attached chiral Au NC. The experiments and fundamental understanding of the photonic spin-Hall effect that governs the coupling between the SPP modes in the Ag NW and the spin states of light have been provided. The distribution of the ellipticity of the (chiral Au NC)–(Ag NW) structure has been investigated by FDTD simulations to analyze the routing direction of the SPPs. The chiroptical response of the chiral Au NC and the photonic spin-Hall effect of the SPPs are utilized to control the coupling of light and valley excitons and route the excitonic emissions with different spins into different directions. The chiroptical response of the (chiral Au NC)–(Ag NW)-on-WS$_2$ structure has been found to play a crucial role in effectively routing the SPPs, while simultaneously enhancing the DCPs of the output emissions. The ability to control SPPs has the potential to improve the emission efficiency and signal integrity in valleytronic devices, thereby enhancing their overall performance and functionality. In the future, such a chiral waveguide with robust directionality will enable efficient routing and switching of quantum information between different components. The chiral waveguide can also be used as a filter to block unwanted signals that may interfere with desired ones.

## Methods

### Synthesis of the chiral Au NCs

The chiral Au NCs were synthesized using a two-step wet-chemistry method. Au nano-octahedrons with an edge size of ~30 nm were first grown by a seed-mediated method[31]. A freshly prepared, ice-cold sodium borohydride solution ($NaBH_4$, 0.02 M, 0.45 mL) was mixed with an aqueous solution composed of gold chloride trihydrate ($HAuCl_4$, 0.01 M, 0.25 mL), cetyltrimethylammonium chloride (CTAC, 0.2 M, 5 mL), and water (4.75 mL) under vigorous stirring. The resultant seed solution was kept at 35 °C for 1 h. Two solutions were prepared for the growth of the Au nano-octahedra. The solutions 1 and 2 were made by mixing CTAC (0.2 M, 20 mL), $HAuCl_4$ (0.01 M, 1 mL), KI (0.001 M, 0.2 mL), ascorbic acid (AA, 0.04 M, 0.88 mL), and water (17.92 mL) sequentially. The formation of the Au nano-octahedrons was initiated by adding the seed solution (0.22 mL) to the growth solution 1, which was followed by gentle shaking for 5 s. The resultant solution (0.22 mL) was then added to the growth solution 2. The obtained mixture solution 2 was mixed by gentle shaking and then left undisturbed at room temperature for 45 min and then centrifuged at $1101 \times g$ with a rotor radius of ~10.94 cm for 10 min. These nano-octahedrons (0.188 mL) were then used as the seeds for the growth of chiral Au NCs in an aqueous solution containing cetyltrimethylammonium bromide (CTAB, 0.1 M, 1 mL), AA (0.2 M, 0.25 mL), $HAuCl_4$ (0.01 M, 0.3 mL), and glutathione (GSH, 2.75 mM, 0.12 mL). The mixture solution turned red after 3 h of storage at 35 °C. The GSH molecules tethered to the Au surface controlled the chirality of the high-Miller-index facets, resulting in chiral Au NCs with four highly curved arms on each facet. The average edge size of the chiral Au NCs is ~150 nm.

### Synthesis of the Ag NWs

The Ag NWs were synthesized through the overgrowth of Ag on Au nanobipyramid (NBP) seeds[32,33]. A freshly prepared and ice-cold $NaBH_4$ (0.01 M, 0.15 mL) was rapidly mixed with an aqueous mixture containing $HAuCl_4$ (0.01 M, 0.125 mL), trisodium citrate (0.01 M, 0.25 mL), and water (9.625 mL) under vigorous stirring. The resultant seed solution was kept at room temperature for 2 h. Subsequently, 0.8 mL of the as-prepared seed solution was introduced into a pre-mixed growth solution consisting of CTAB (0.1 M, 40 mL), $HAuCl_4$ (0.01 M, 2 mL), silver nitrate ($AgNO_3$, 0.02 M, 0.2 mL), HCl (1 M, 0.8 mL), and AA (0.2 M, 0.16 mL). After being stirred for 2 min, the mixture was left undisturbed overnight at 35 °C. The Au NBPs were centrifuged at $5992 \times g$ with a rotor radius of ~10.94 cm for 10 min and redispersed in 10 mM CTAB solution for subsequent use. The diameter of the Ag NWs was determined by the waist diameter of the Au NBPs. For the NBP samples used in our experiments, the extinction value per 0.5 cm at the longitudinal plasmon peak of 800 nm was 1.7. The Au NBP solution (40 mL) was centrifuged at $4403 \times g$ with a rotor radius of ~10.94 cm for 8 min. The precipitate was redispersed in cetyltrimethylammonium chloride (CTAC, 0.08 M, 30 mL), followed by subsequent addition and mixing of $AgNO_3$ (0.02 M, 4 mL) and AA (0.2 M, 2 mL). The mixture solution was placed in an air-bath shaker (60 °C, 180 rpm) and kept for 6 h. The resultant sample was centrifuged at $1957 \times g$ with a rotor radius of ~10.94 cm for 10 min. The precipitate was redispersed in CTAB (0.03 M, 30 mL) overnight for sedimentation. The supernatant was composed of irregular Ag nanoparticle impurities, and the sediment was composed of pure Ag nanorods. The precipitate was redispersed in CTAB (0.08 M) for another overgrowth process to grow longer Ag nanorods. The volumes of the $AgNO_3$ solution (0.02 M) and AA solution (0.2 M) in each overgrowth cycle for the Ag NW sample were 8 mL and 4 mL, respectively. The overgrowth process was repeated several times until the expected length was reached. The Ag NW precipitate was finally redispersed in CTAB (0.05 M, 30 mL) for long-term storage. The transmission electron microscopy (TEM) image of the synthesized Au NBP seeds and the SEM image of the Ag NWs with an average length of 8 µm are shown in Supplementary Fig. 1a, b, respectively.

### Fabrication of the (chiral Au NC)−(Ag NW) and (chiral Au NC)−(Ag NW)-on-$WS_2$ structures

The diluted Ag NW solution was initially deposited onto a Si/$SiO_2$ substrate and left undisturbed for 10 s. The substrate was then blown dry with nitrogen gas. The chiral Au NCs were then deposited onto the same substrate and left undisturbed for 20 s followed by the same drying procedure. The chiral Au NCs were adhered to the surface of the Ag NWs. $WS_2$ monolayers were grown on Si/$SiO_2$ substrates at temperatures of 1170–1200 °C by physical vapor deposition[28]. The grown $WS_2$ monolayers had high crystallinity and large areas of ~1 $mm^2$. The (chiral Au NC)−(Ag NW)-on-$WS_2$ structures were fabricated by a transfer process. The diluted Ag NW solution was initially deposited onto a polydimethylsiloxane (PDMS) substrate. The PDMS substrate was then baked on a hot plate at 35 °C for 2 min and blown dry with nitrogen gas. The chiral Au NCs were then deposited onto the same substrate and left undisturbed for 4 min. The substrate was blown dry again with nitrogen gas. The ideal (chiral Au NC)−(Ag NW) structures deposited on the PDMS substrate were transferred onto a piece of $WS_2$ monolayer after close contact between the surface of the PDMS substrate and the $WS_2$ monolayer at 50 °C for 20 min. The PDMS film was then peeled off from the Si/$SiO_2$ substrate.

### Single-particle optical measurements and characterization

Backward dark-field scattering measurements were carried out on a home-built optical microscope. An upright optical microscope (Olympus, BX53M) was integrated with a quartz-tungsten-halogen lamp (100 W) and a digital color camera (Olympus, DP73). A 100× dark-field air objective (Olympus, numerical aperture: 0.9) was used. The incident light from the halogen lamp passed through a dark-field objective to form an annular light beam. The annular light was focused onto the sample at an oblique incident angle. The scattered light was collected by the same objective in the case of backward scattering and then directed toward a digital color camera. The dark-field scattering images of the chiral Au NCs and Ag NWs could thus be captured. The measurements of scattering spectra were conducted using the optical microscope equipped with a monochromator (Acton, SpectraPro 2360i) and a cooled CCD camera (Princeton Instruments, Pixis 400) maintained at a temperature of −70 °C. Circularly polarized excitation was realized by a linear polarizer and a quarter-wave plate (WPA4420-450-650, Union Optic) with a working spectral range of 450–650 nm. The polarization handedness convention used in this work was such that the RCP and LCP vectors rotate clockwise and counterclockwise along the propagation axis, respectively. The scattering $g$-factor is defined as $g\text{-factor} = 2 \times (S_{LCP} - S_{RCP})/(S_{LCP} + S_{RCP})$, with $S_{LCP}$ and $S_{RCP}$ corresponding to the scattering spectra under LCP and RCP light, respectively. Extinction spectra were measured on a PerkinElmer Lambda 950 ultraviolet/visible/near-infrared spectrophotometer. SEM imaging was carried out on an FEI QF400 field-emission scanning electron microscope operated at 20 kV. TEM imaging was carried out on an FEI TEM TS12 operated at 120 kV.

### Surface plasmon polariton excitation and detection

The experimental setup for the measurements of the SPP outputs of the NW was based on a laser light source, a plasmonic waveguide, an optical microscope, and a digital color camera. A 633-nm laser beam was used as the excitation source. The incident laser light was focused on the chiral Au NC. The incident light was scattered at the asymmetrical positions on the surface of the NW, causing the momentum component of light to align with that of the SPPs. The illumination at the chiral Au NC, therefore, led to SPP outputs at the ends of the NW. The scattered light was collected with a 100× objective and then directed to the color camera. The image of the scattered light was taken to assist in visualizing the spatial distribution and intensity of the SPP outputs from the NW.

## Preparation of the (chiral Au NC)@SiO$_2$ nanoparticles

The (chiral Au NC)@SiO$_2$ core@shell nanoparticles were synthesized by the growth of a mesostructured SiO$_2$ shell onto the chiral Au NCs through the hydrolysis of tetraethyl orthosilicate (TEOS). The chiral Au NC sample was washed to remove extra CTAB and redispersed in water (1 mL). NaOH (0.1 M, 10 μL) and CTAB (0.1 M, 10 μL) solutions were then added. A SiO$_2$ precursor was prepared by mixing TEOS (1 mL) with ethanol (9 mL). Under continuous stirring, the precursor solution (10 μL) was added dropwise to the chiral Au NC dispersion every 20 min. After the addition of the precursor solution in 1 h (a total of 30 μL of the precursor solution added), stirring was continued for 5 h to allow for the hydrolysis and formation of a mesostructured SiO$_2$ shell. The obtained nanoparticle sample was finally washed twice and redispersed in ethanol.

## Preparation of the (chiral Au NC)–fluorophore nanoparticles

The (chiral Au NC)–fluorophore nanoparticles were synthesized with a modified procedure to incorporate the R640 fluorophore into the mesostructured SiO$_2$ shell. The process began identically, with the washing and redispersion of the chiral Au NCs and the addition of the same NaOH and CTAB solutions. The key difference was in the preparation of the precursor solution. R640 (50 mg) was first dissolved in ethanol (5 mL), and the resultant solution was then mixed with TEOS (1 mL) and ethanol (4 mL) to form a fluorophore-containing SiO$_2$ precursor. The same protocol (3 μL every 20 min for 1 h) was then executed, which was followed by stirring for a total time of 5 h. The SiO$_2$ shell formed on the surface of the chiral Au NCs with R640 molecules embedded. The final nanoparticles were washed twice and redispersed in water.

## PL spectroscopy

The experimental setup for the measurements of PL spectra was based on a laser light source, a set of polarizers, a long-pass filter, an optical microscope, a monochromator, and a cooled CCD camera. Two lasers at the wavelengths of 514 nm and 594 nm were employed for PL excitation. A linear polarizer was inserted between the laser output port and the beam splitter for the control of the linear polarization direction of the laser light. A 593 nm long-pass filter (FF01-593/LP-25, AVR Optics) and a 550 nm long-pass filter (FELH0550, Thorlabs) were placed in front of the spectrometer to block the 594 nm and 514 nm excitation laser light, respectively. An additional set of optical polarizers, including a linear polarizer and a broadband quarter-wave plate, was inserted between the beam splitter and the long-pass filter for the detection of polarization-resolved PL spectra. PL and polarization-resolved PL spectra were measured by the monochromator and the cooled CCD camera.

## Electrodynamic simulations

The electromagnetic simulations were performed using FDTD Solutions 2020 R2 (Lumerical) and COMSOL Multiphysics 6.3. During FDTD simulations, a (chiral Au NC)–(Ag NW) structure was placed on a substrate. A Gaussian beam was launched at the central position of the chiral Au NC along the $-z$ direction. An etched cube model was used to emulate the morphology of the actual chiral Au NC as closely as possible. The Ag NW was modeled as a cylinder with a length of 8 μm. The gap distance between the chiral Au NC and the Ag NW was set at 2 nm. The WS$_2$ monolayer with a thickness of 1 nm was modeled, with the thickness of the SiO$_2$ substrate set at 740 nm to occupy the simulation region below the monolayer. In the simulations, a mesh size of 2 nm was used for the chiral Au NC, 1 nm for the WS$_2$ monolayer, and 3 nm for the Ag NW. These mesh sizes were chosen to balance the computational accuracy and efficiency. The dielectric functions of Au and Ag were taken from Johnson and Christy's data. The dielectric function of the WS$_2$ monolayer with a thickness of 1 nm was calculated by fitting the experimental data in a

previous work[34]. The refractive index of the SiO$_2$ substrate was set at 1.45. CPL was produced when the two orthogonal electric field component vectors were of equal magnitude and out of phase by 90°. The phase of the $x$-polarized plane wave was fixed at 0°. A positive phase difference of +90° between the $y$- and $x$-polarized plane waves gave LCP light. A negative phase difference of −90° gave RCP light. For the simulations of decay rates, 633 nm circular dipole sources were placed below the (chiral Au NC)–(Ag NW) structure. During COMSOL simulations on the plasmon modes of the Ag NW and the overlapping intensities, a perfectly matched layer was applied as the boundary to absorb the outgoing radiation. The refractive index of the air was set at 1.00.

## Data availability

The data generated in this study are provided in the Supplementary Information/Source data file. Source data are provided with this paper.

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

## Acknowledgements

J.F.W. acknowledges support from the Research Grants Council of Hong Kong (ANR/RGC, A-CUHK404/21 and GRF, 14303124) and the Direct Grant of The Chinese University of Hong Kong (Project Code: 4053670). Y.L.C. acknowledges being supported by the China Postdoctoral Science Foundation under Grant Number 2024M753075.

## Author contributions

J.F.W. conceived the project. Y.L.C. and Y.C contributed equally to the device fabrication, optical measurements, and theoretical calculations. Y.L.C. performed the simulations. Y.N.F carried out the chemical synthesis of the Ag NWs. R.Q.A. and X.M.C. prepared the TMDC monolayers. J.F.W. and Y.L.C. co-wrote the manuscript. J.F.W. and X.L.Z. supervised the project. All authors discussed the results and commented on the manuscript.

## Competing interests

The authors declare no competing interests.
