## [Peer Review File · Nature Communications]

Photonic spin-Hall effect in chiral plasmonic assemblies

Corresponding Author: Professor Jianfang Wang

Version 0:

Reviewer comments:

Reviewer #1

(Remarks to the Author)

This work reports an assembly system of chiral Au NCs onto the surface of Ag NWs, and demonstrate the routing effect of SPPs modulated by the chirality of attached Au NCs under excitation of linearly polarized light. Furthermore, the Au NC-Ag NW assembly on WS₂ monolayer induce directionally emissions of valley-polarized excitons and enhance the degree of valley polarization. The underlying mechanism understanding is further explained through numerical simulations. The optical chirality and spin manipulation are intriguing physical properties and have drawn intense attention in many research fields. This work presents an interesting assembly structure and clear experimental results. I think it could be accepted for publication. While some concerns listed below should be addressed.

1. In this work, the experimental results of left-handed and right-handed devices are compared with bare Ag NWs. Why not using an assembly of achiral nanoparticle-NW hybrid? The coupling of nanoparticle of Ag NWs may lead some effects exist in chiral Au NC–NW hybrid, while not exist in bare Ag NWs.
2. In Fig. 3f, the difference between RO and LO is very small. It may lead to a g-factor even smaller than that of the dashed lines in Fig. 3e of bare Ag NW. In the experiment, what reason would affect the chiral responses in the hybrid system?
3. In Fig. 4, the DCP enhancement of left-handed and right-handed devices is evaluated based on the AgNW-on-WS₂ sample. How about the result if a sample of achiral AuNC-AgNW-on-WS₂ was used? Does the coupling of nanoparticle and nanowire influence the chiral responses?
4. The 594-nm and 514-nm excitations lead to different influence for the left-handed and right-handed devices. What is the detailed reason?
5. Could the distance between chiral Au NCs and Ag NWs be controlled in the experiment? This distance may influence the plasmon coupling and then the SPP excitation in the nanowires. Meanwhile, the space distance between plasmonic structure and WS₂ monolayer may affect the exciton PL and emission dynamics.

Reviewer #2

(Remarks to the Author)

Directional light splitting based on the photonic spin-Hall effect is demanding for the development of spin-dependent optical elements but still greatly missing. Here we report on the routing of surface plasmon polaritons (SPPs) by use of chiral Au nanocubes (NCs). Experiments and theoretical modeling have been performed to understand the photonic spin-Hall effect in an Ag nanowire (NW) under excitation of circularly polarized light. The SPPs in the Ag NW can be modulated by the attached chiral Au NCs with opposite chiralities under excitation of linearly polarized light. Hybrid structures are further constructed out of (chiral Au NC)-(Ag NW) assemblies and transition metal dichalcogenide monolayers to selectively induce directional emissions of valley-polarized excitons and enhance the degree of valley polarization. The research group has good experimental equipment and proficient operational skills. Many drawing details expressions are need modified.

1. In the caption of Figure 1 (e), directly using CPL to describe the experimental design lacks guidance. It is suggested to make the following two modifications:

- 1) supplement the previous text to clarify that CPL has two chirality, namely LCP/RCP.
- 2) clarify in the relevant discussion of Figure 1 (e) that the experiment was conducted separately by LCP and RCP

2. Taking Figure 2 (b) as an example, it is recommended to use two different colors, red and blue, to distinguish between LCP and RCP light sources with different chirality.
3. The article does not discuss the chirality of gold nanocubes. How is their chirality determined through SEM images? I suggest the author can clearly indicate or supplement the synthesis pathway, or provide corresponding chiral optical effects that can be distinguished. It is recommended that the author refer to the judgment methods of relevant research on the synthesis of this particle for discussion and supplementation
4. Supporting Figure S8(a) suggest placing the coordinate axis at the front to help readers read and understand.
5. Will this quadrilateral structure have resonance problems like previous studies ? in FigureS17
ref : L. Shan, F. Zhang,J. Ren et al. Large Purcell enhancement with nanoscale non-reciprocal photon. transmission in chiral gap-plasmon-emitter systems. Opt Express, 2020,28(23): 33890-33899
6. The structure of WS2 and silver nanowires can also cause directional propagation when incident by CPL. Will linearly polarized light incident in the article also have a similar effect? I hope the author can further discuss this issue. If linearly polarized light directly irradiates the WS2 line structure without causing directional propagation, it directly proves the optical effect of gold nanostructures; If it will lead to directional propagation, the author needs to supplement the discussion on how to avoid the influence of silver nanowires on the experimental results when designing the experiment (such as removing only gold nanoparticles to see if the propagation direction will be affected).
7. Can we discuss the impact of geometric positions? How to control the relative position of gold and silver in the experiment? If the experiment cannot be controlled, should we consider adding relevant simulations for theoretical calculations?
8. In this study, the optical properties of gold nanocubes were limited to the conversion of linearly polarized light. The article also mentioned the enhancement effect of the structure's composition on electromagnetic fields. Can Raman and other optical properties be further studied?
- 9 figure 1 D-hand and L-hand's color bar have different value, why is the same value?

Version 1:

Reviewer comments:

Reviewer #1

(Remarks to the Author)

The revised version has addressed the comments, and the manuscript has been improved accordingly. I think it could be accepted for publication now.

Reviewer #2

(Remarks to the Author)

Directional light splitting relying on the photonic spin-Hall effect is crucial for spin-dependent optical elements yet remains largely unexplored, and this work reports the routing of surface plasmon polaritons (SPPs) using chiral Au nanocubes (NCs). Experiments and theoretical modeling are conducted to investigate the photonic spin-Hall effect in Ag nanowires (NWs) under circularly polarized light excitation, and linearly polarized light-driven SPPs in Ag NWs can be modulated by attached chiral Au NCs with opposite chiralities. Hybrid structures composed of chiral Au NC-Ag NW assemblies and transition metal dichalcogenide monolayers are constructed to selectively trigger directional emissions of valley-polarized excitons and boost valley polarization degree, whose routing mechanism is clarified via numerical simulations and holds great significance for valleytronic circuit development.

The quality of the data is technically sound and presented in sufficient detail.

The level of support for the conclusions are sufficiently strong evidence is provided for the authors' claims and all appropriate controls have been included.

these results will be important to the field and advance understanding in a way that will move the field forward.

Journal: *Nature Communications*

Manuscript ID: NCOMMS-25-63656

Title: “Chiral plasmonic nanoparticle-directed photonic spin-Hall effect”

Authors: Yilin Chen, Yang Chen, Yini Fang, Ruoqi Ai, Ximin Cui, Xiaolu Zhuo, Jianfang Wang

Text coding: *black italic, reviewers' comments*; black normal, authors' response; **red normal, changes made in the manuscript**

Response to Reviewer #1

Comments: *This work reports an assembly system of chiral Au NCs onto the surface of Ag NWs, and demonstrate the routing effect of SPPs modulated by the chirality of attached Au NCs under excitation of linearly polarized light. Furthermore, the Au NC-Ag NW assembly on WS2 monolayer induce directionally emissions of valley-polarized excitons and enhance the degree of valley polarization. The underlying mechanism understanding is further explained through numerical simulations. The optical chirality and spin manipulation are intriguing physical properties and have drawn intense attention in many research fields. This work presents an interesting assembly structure and clear experimental results. I think it could be accepted for publication. While some concerns listed below should be addressed.*

Response: We thank this reviewer for the effort made to evaluate our work, the positive comments, and the valuable suggestions.

Question 1: *In this work, the experimental results of left-handed and right-handed devices are compared with bare Ag NWs. Why not using an assembly of achiral nanoparticle-NW hybrid? The coupling of nanoparticle of Ag NWs may lead some effects exist in chiral Au NC-NW hybrid, while not exist in bare Ag NWs.*

Response: We thank the reviewer for this valuable suggestion. The reason for investigating the chiral devices is to demonstrate the effect of chirality. We agree that comparing the results with those of achiral hybrids is essential. The achiral hybrids can exhibit directional propagation only under circularly polarized light (CPL), whereas the chiral hybrids exhibit robust directional propagation under linearly polarized light.

We have added new results of the surface plasmon polariton (SPP) propagation in a Ag nanowire (NW) with an attached Au nanocube (NC) under the excitation of left-handed and right-handed circularly polarized (LCP and RCP) laser light in Supplementary Fig. 7. The edge length of the Au NC was ~130 nm. The 633 nm LCP and RCP laser light were directionally coupled toward the left and right outputs (LO and RO) of the NW, respectively. The output intensities for 3 (Au NC)-NW structures showed similar results. We define the reversed spin-Hall effect where under the excitation of light along the $-z$ direction, LCP and RCP exciting light propagate toward the LO and RO, respectively. The propagation directions for LCP and RCP light in the achiral hybrids are opposite to those in the chiral hybrids. Such a reversal highlights the role of plasmon coupling in spin-dependent routing. The coupling of the

nanoparticle and the NW can modify the direction of the localized energy flow in the gap region between the nanoparticle and the NW. The simulated results will be discussed later.

Supplementary Fig. 7 | Reversed photonic spin-Hall effect in the (Au NC)–(Ag NW) structures. **a** SEM images of a (Au NC)–(Ag NW) structure. The Au NC is located at the central position of the Ag NW. The edge length of the Au NC is ~ 130 nm. **b** Pseudocolor dark-field microscopy image of the (Au NC)–NW structure (i) and pseudocolor images for the illumination of LCP (ii) and RCP (iii) laser light. The LCP and RCP laser light are directionally coupled toward the LO (red box) and RO (yellow box) of the NW, respectively. **c** Normalized intensities of the LO and RO of the SPP propagation in the (Au NC)–NW structures under the excitation of LCP and RCP light. The data were collected and averaged from 3 (Au NC)–NW structures, respectively. Normal photonic spin-Hall effect in Fig. 1c is defined as the scenario in which under the excitation of CPL along the $-z$ direction, LCP- and RCP-excited SPPs propagate toward the RO and LO of the NW, respectively. In contrast, the reversed spin-Hall effect in (b) describes the flipped case where under the same incident CPL along the $-z$ direction, LCP- and RCP-excited SPPs propagate toward the LO and RO, respectively.

The SPP propagation in the (Au NC)–(Ag NW) structures under the excitation of linearly polarized light has been added in Supplementary Fig. 8. Under linear polarization excitation, the output intensity difference between the LO and RO of the NW was insignificant. The output intensities for 4 (Au NC)–NW structures showed similar results (Supplementary Fig. 8c). These experiments confirm that the strong directional output observed in the (chiral Au NC)–NW structures under linearly polarized light arises from the combined effect of chiroptical responses and the photonic spin-Hall effect.

Supplementary Fig. 8 | Surface plasmon propagation in the (Au NC)-(Ag NW) structures under the excitation of linearly polarized light. **a** SEM images of a (Au NC)-(Ag NW) structure. The Au NC is located at the central position of the Ag NW. **b** Pseudocolor dark-field microscopy image of the (Au NC)-NW structure (i) and pseudocolor image for the illumination of linearly polarized laser light (ii). The output intensity difference between the RO (yellow box) and LO (red box) of the NW is insignificant. **c** Normalized intensities of the LO and RO of the SPP propagation in the (Au NC)-NW structures under the excitation of linearly polarized laser light. The data were collected and averaged from 4 (Au NC)-NW structures.

The normalized intensity and the directionality of the SPP outputs for the (achiral Au NC)-NW structures under the excitation of linearly polarized light have been included in Fig. 1f, g.

Fig. 1 | Photonic spin-Hall effect in the (chiral Au NC)-NW structures. **a, b** Schematics and SEM images of **f, g** Normalized intensities of the LO and RO (**f**) and directionality of the SPP propagation (**g**) in the (D-/L-handed chiral Au NC)-NW structures and (achiral Au

NC)–NW structures under the excitation of linearly polarized light. The data were collected from 4 (D-/L-handed chiral Au NC)–NW structures and 4 (achiral Au NC)–NW structures, respectively.

To understand the reversed spin-Hall effect in achiral structures, the distributions of the electric field enhancement for the (Au NC)–(Ag NW) structure under the excitation of polarized dipole sources were simulated (Supplementary Fig. 12a–c). LCP and RCP dipoles were respectively routed along the $-x$ and $+x$ directions. Because the corners of Au NCs synthesized in experiments are often rounded, we considered in the simulations Au NCs with their 6 corners rounded in different degrees. The sign of the directionality was found to remain unchanged, and the magnitude of the directionality increased when the attached nanoparticle was changed from a gold NC with sharp and right-angle corners to NCs with rounded corners. Our simulation results explained the reversed spin-Hall effect observed in the (Au NC)–(Ag NW) structures. Under linear polarization excitation, no directional propagation was observed for achiral structures (Supplementary Fig. 12a).

Supplementary Fig. 12 | Simulations of the electric field of the (Au NC)–(Ag NW) structures. **a** Distributions of the electric field enhancement in an x - y plane monitor located at the bottom surface of the Ag NW in the (Au NC)–(Ag NW) structure under the excitation of circularly and linearly polarized dipole sources. For circularly polarized dipoles, SPPs propagated along the $-x$ direction for the LCP dipole, whereas SPPs propagated along the $+x$ direction for the RCP dipole. For linearly polarized dipoles, no directional propagation was observed in the achiral structures. **b, c** Reversed photonic spin-Hall effect for the (Au NC with rounded corners)–(Ag NW) structures under the excitation of circularly polarized dipole sources. The corner-rounded Au NCs with fillet radii of 20 nm and 50 nm are denoted as R20 (**b**) and R50 Au NC (**c**), respectively. The sign of the directionality remained unchanged relative to (**a**), but the magnitude of the directionality increased as the corners of the attached

NC became more rounded. **d** Normal spin-Hall effect in the (Au NC dimer)–(Ag NW) structure under the excitation of circularly polarized dipole sources. LCP and RCP excitations were directed toward the RO and LO of the NW, respectively. The position of the x - y plane monitor is identical in (a–d). The left panel schematically shows the cross-sectional configurations of the achiral Au NCs attached to the surface of the Ag NW.

For the (Au NC dimer)–(Ag NW) structure (Supplementary Fig. 12d), LCP and RCP dipoles propagated along the $+x$ and $-x$ directions, respectively. This reversal between the normal and reversed photonic spin-Hall effects can be attributed to a sign change in the z -component of the Poynting vector (P_z) induced by plasmon coupling between the nanoparticles and the NW. For LCP light incident along the $-z$ direction, the plasmonic interaction in the gap region can flip the sign of P_z , as evidenced by its distribution along a line across the nanoparticle–NW gap (Supplementary Fig. 19). P_z was negative (pointing in the $-z$ direction) for structures with an attached Au NC dimer or D-/L-handed chiral Au NC, whereas P_z was positive (pointing in the $+z$ direction) for those with R50 or R20 Au NCs. Since the opposite propagating direction of light in the coupling fields gives rise to opposite SPP coupling direction, this sign reversal of P_z from negative to positive accounts for the transition from the normal to the reversed spin-Hall effect. This interpretation is consistent with the simulation results under the excitation of circularly polarized dipole sources in Supplementary Figs. 10, 12 and Fig. 2a.

Supplementary Fig. 19 | Simulations of the z -component of the Poynting vector P_z under LCP Gaussian beam. **a** Schematic of the (R50 Au NC)–(Ag NW) structure and the distribution of P_z in an x - y plane monitor. The red dashed line represents the position of the R50 Au NC. **b–f** Distributions of P_z in the line at the gap between the plasmonic nanoparticle and the Ag NW as shown in the black dashed line of (a). The plasmonic nanoparticles are R50 Au NC (b), R20 Au NC (c), Au NC dimer (d), D-handed chiral Au NC (e), and L-handed chiral Au NC (f), respectively. The directions of P_z in the (R50 Au NC)–NW and (R20 Au NC)–NW structures (b, c) are along the $+z$ direction, which are opposite to those of the cases with the Au

NC dimer and the chiral Au NCs (d–f).

We have added the experimental results of the normal photonic spin-Hall effect for a (Au NC dimer)–(Ag NW) structure under the excitation of LCP/RCP light in Supplementary Fig. 9. The LCP and RCP laser light were directionally coupled toward the RO and LO of the NW, respectively.

Supplementary Fig. 9 | Photonic spin-Hall effect in a (Au NC dimer)–(Ag NW) structure. **a** SEM images of a (Au NC dimer)–(Ag NW) structure. The Au NC dimer is located at the central position of the Ag NW. **b** Pseudocolor dark-field microscopy image of the (Au NC dimer)–NW structure (i) and pseudocolor images for the illumination of LCP (ii) and RCP (iii) laser light. The LCP and RCP laser light were directionally coupled toward the RO (yellow box) and LO (red box) of the NW, respectively.

We have added the discussion about the SPP propagation for achiral structures at the end of the paragraph right before the section of “Theoretical demonstration of the photonic spin-Hall effect”.

“..... In addition, only the light spot from the LO could be detected when the attached L-handed chiral Au NC was spatially rotated with respect to the ideal devices (Supplementary Fig. 6b, c), which indicates the maintenance of the chirality-dependent directional propagation in the structure. To further make comparison with the results from the chiral structures, the SPP propagation in achiral structures was investigated (Fig. 1f, g and Supplementary Figs. 7–9). An achiral Au NC was assembled onto the surface of the Ag NW. LCP and RCP exciting light were directionally coupled to the LO and RO of the NW, respectively, which is referred to as a reversed photonic spin-Hall effect. Such a reversal highlights the role of plasmon coupling in spin-dependent routing. Under the excitation of linearly polarized light, the intensity difference between the LO and RO was negligible. The directional SPP propagation observed in the (chiral Au NC)–NW structures under linearly polarized light therefore originates from their chiroptical responses and the photonic spin-Hall effect.”

We have added the discussion about Supplementary Fig. 12 at the end of the paragraph right before Fig. 2.

“..... The FDTD simulations of the (chiral Au NC)–(faceted pentagonal Ag NW) structures suggest that the spin-direction locking effect and the directional SPP propagations induced by the chirality of the attached chiral nanoparticle remain robust (Supplementary Fig. 11b–d). The simulations of the achiral hybrid structures demonstrated that the direction of SPP propagation

under the excitation of CPL can be tuned by the plasmon coupling of the NW with different attached nanoparticles (Supplementary Fig. 12).”

We have added the discussion about Supplementary Fig. 19 in the paragraph right after Fig. 2.

“..... The coupling efficiency of the plasmon modes in the NW and CPL is therefore determined by the polarization and the distribution of the localized electromagnetic field enhancement on the surface of the NW. The coupling of the nanoparticle and the NW modifies the directions of the Poynting vectors in the gap between the nanoparticle and the NW. Further analysis of the z -component of the Poynting vector (P_z) in Supplementary Fig. 19 reveals that the sign reversal of P_z from negative to positive accounts for the transition from the normal to the reversed spin-Hall effect. The theoretically demonstrated photonic spin-Hall effect can be used to explain

We have also added discussion after the caption of Supplementary Fig. 19.

“The reversal between the normal and reversed photonic spin-Hall effects can be attributed to the sign change of P_z induced by the plasmon coupling between the nanoparticle and the NW. For LCP light incident along the $-z$ direction, the plasmonic interaction in the gap region flips the sign of P_z , as evidenced by its distribution along a line across the nanoparticle–NW gap (Supplementary Fig. 19). P_z is negative (pointing in the $-z$ direction) for structures with an attached Au NC dimer or D-/L-handed chiral Au NC, whereas P_z is positive (pointing in the $+z$ direction) for those with the R50 or R20 Au NC. Since the opposite propagating directions of light in the coupling fields give rise to opposite SPP coupling directions, this sign reversal of P_z from negative to positive accounts for the transition from the normal to the reversed spin-Hall effect. This interpretation is consistent with the simulation results under the excitation of circularly polarized dipole sources in Supplementary Figs. 10, 12 and Fig. 2a.”

Question 2: *In Fig. 3f, the difference between RO and LO is very small. It may lead to a g -factor even smaller than that of the dashed lines in Fig. 3e of bare Ag NW. In the experiment, what reason would affect the chiral responses in the hybrid system?*

Response: We thank this reviewer for the valuable comment and question. The small difference between the RO and LO in Fig. 3f is attributed to the weak chiroptical response of the D-handed structure at 514 nm. This is corroborated by the scattering dissymmetry factor (g -factor) measurements of the chiral structures presented in Supplementary Fig. 27. The intensity of the scattering g -factor for the D-handed structure at 514 nm is significantly lower than that at 594 nm.

The corresponding discussion has been expanded and presented more clearly in the second paragraph after Fig. 3.

“..... In contrast, the scattering g -factor of the attached L-handed chiral Au NC is negative. The sign and magnitude of the scattering g -factor reflect the handedness and strength of the chiroptical response, which governs the spin-selective coupling of linearly polarized incident light into SPPs. A larger positive g -factor at a given wavelength indicates stronger preferential scattering of LCP light, which leads to more pronounced directional propagation of SPPs

toward the RO of the NW. A negative scattering g -factor implies dominant RCP scattering and directional outputs toward the LO. The directionalities under the excitation of 594 nm and 514 nm laser light in

Question 3: *In Fig. 4, the DCP enhancement of left-handed and right-handed devices is evaluated based on the AgNW-on-WS₂ sample. How about the result if a sample of achiral AuNC-AgNW-on-WS₂ was used? Does the coupling of nanoparticle and nanowire influence the chiral responses?*

Response: We thank the reviewer for raising this important point. We have performed the suggested measurements using achiral structures. The results have been added in Supplementary Figs. 30, 31. When a single Au NC is coupled to the Ag NW on WS₂ monolayer (Supplementary Fig. 30), the photoluminescence (PL) intensities from the LO and RO show negligible difference, and the polarization-resolved PL spectra reveal minimal degrees of circular polarization (DCPs) under either 594 nm or 514 nm excitation. Both directionality and DCP of the outputs under linear polarization excitation are markedly weaker than those observed for the (chiral Au NC)–NW devices as shown in Figs. 3, 4. Even with the stronger electric field enhancement induced by the (achiral Au NC dimer)–NW structure (Supplementary Fig. 31), no directional propagation emerges. These results confirm that the directional circular polarization emissions in the chiral structures originate from the chiroptical response of the attached chiral nanoparticle.

Supplementary Fig. 30 | PL spectra and polarization-resolved PL spectra of the outputs of the (Au NC)-(Ag NW)-on-WS₂ structure. a SEM images of a (Au NC)-(Ag NW)-on-WS₂ structure. **b** PL spectra of the LO and RO under the excitation of 594 nm linearly polarized laser light. **c, d** Polarization-resolved PL spectra of the LO (**c**) and RO (**d**) under the excitation of 594 nm linearly polarized laser light. **e** PL spectra of the LO and RO under the excitation of 514 nm linearly polarized laser light. **f, g** Polarization-resolved PL spectra of the LO (**f**) and RO (**g**) under the excitation of 514 nm linearly polarized laser light.

Supplementary Fig. 31 | PL spectra and polarization-resolved PL spectra of the outputs of the (Au NC dimer)-(Ag NW)-on-WS₂ structure. a SEM images of a (Au NC dimer)-(Ag NW)-on-WS₂ structure. **b** PL spectra of the LO and RO under the excitation of 594 nm linearly polarized laser light. **c, d** Polarization-resolved PL spectra of the LO (**c**) and RO (**d**) under the excitation of 594 nm linearly polarized laser light. **e** PL spectra of the LO and RO under the excitation of 514 nm linearly polarized laser light. **f, g** Polarization-resolved PL spectra of the LO (**f**) and RO (**d**) under the excitation of 514 nm linearly polarized laser light.

The corresponding discussion has been added to the end of the paragraph right before Fig. 4.

“..... The absolute DCP values of the ROs are higher than those of the LOs for both the L-handed and D-handed structures. To further clarify the essential role of chiral plasmonic nanoparticles, we conducted a series of control experiments using a bare (Ag NW)-on-WS₂ structure without any attached nanoparticle and (achiral Au NC)-NW-on-WS₂ structures (Supplementary Figs. 29–31). In all control cases, whether no particle, a single achiral Au NC, or an achiral Au NC dimer was attached, the PL intensities from the LO and RO showed negligible differences under linear polarization excitation, and the DCP remained close to zero. These results confirm that the chiral Au nanoparticle is essential for achieving the observed

directional routing of PL. In contrast, achiral hybrids or the NW alone do not support directional output under the same excitation geometry.”

Question 4: *The 594-nm and 514-nm excitations lead to different influence for the left-handed and right-handed devices. What is the detailed reason?*

Response: We thank the reviewer for raising this insightful question. The distinct effects of 594 and 514 nm excitation originate from the peak wavelength of the chiroptical resonances in the hybrid structures. The scattering g -factor of the D-handed device indicates its chiroptical response, which shows the strongest g -factor intensity at the wavelength of 593 nm (Supplementary Fig. 27). When the 594-nm excitation is on resonance with this chiroptical response, the chiral structure generates a field with a higher degree of circular polarization and enables a directional SPP propagation observed in Fig. 3. In contrast, the 514-nm excitation is off-resonance, leading to a much weaker chiroptical response and consequently less effective linear-to-elliptical polarization conversion. The diminished magnitude of ellipticity under the excitation of 514 nm linearly polarized light results in a symmetric SPP propagation.

The corresponding discussion has been presented more clearly in the second paragraph after Fig. 3.

“..... The variations in D_{594} and D_{514} are correlated with the g_{594} and g_{514} values of the attached chiral Au NCs, respectively. The correlation stems from the chiral plasmonic resonance. The scattering g -factor peak at the wavelength of 593 nm indicates strong chiroptical on-resonance with 594 nm excitation. The 594 nm linear polarization excitation therefore efficiently enables a spin-selective coupling into directional SPPs, yielding a high D_{594} . In contrast, the much smaller g -factor at 514 nm (g_{514}) reflects off-resonant chiroptical interaction. This results in a less efficient linear-to-elliptical polarization conversion and a trivial directionality (D_{514}). The higher $|g_{514}|$ value than the $|g_{594}|$ value for the L-handed chiral Au NC results in higher $|D_{514}|$ than $|D_{594}|$ (Fig. 3j).....”

Question 5: *Could the distance between chiral Au NCs and Ag NWs be controlled in the experiment? This distance may influence the plasmon coupling and then the SPP excitation in the nanowires. Meanwhile, the space distance between plasmonic structure and WS₂ monolayer may affect the exciton PL and emission dynamics.*

Response: We thank the reviewer for this excellent question. The response is divided into two parts.

(1) Regarding the question about the distance between the chiral Au NC and the Ag NW, we have added the SPP propagation results for (chiral Au NC)@SiO₂-NW structures in Supplementary Figs. 23, 24. We separated the chiral Au NC and the Ag NW by coating the chiral Au NC with a 15-nm SiO₂ shell (Supplementary Fig. 23). LCP and RCP were respectively routed toward the LO and RO of the NW. The normalized intensities were collected and averaged from 3 (L-handed chiral Au NC)@SiO₂-NW structures under LCP/RCP excitation (Supplementary Fig. 23e). Under linear polarization excitation, the SPPs coupled preferentially to the RO for the (L-handed chiral Au NC)@SiO₂-NW devices, while

for the D-handed devices, they coupled to the LO (Supplementary Fig. 23b, d). The normalized intensities of the LO and RO and directionalities in the (L-/D-handed chiral Au NC)@SiO₂-NW structures under the excitation of linearly polarized light are shown in Supplementary Fig. 23f, g. The directionality shows a reversal compared to the results of the (chiral Au NC)-NW structures. The distance between the chiral Au NC and the Ag NW is a critical parameter that can significantly modulate the plasmon coupling between the nanoparticle and the NW.

Supplementary Fig. 23 | SPP propagation in the (chiral Au NC)@SiO₂-(Ag NW) structures. **a** SEM images of an (L-handed chiral Au NC)@SiO₂-(Ag NW) structure. **b** Pseudocolor dark-field microscopy image of the (L-handed chiral Au NC)@SiO₂-NW structure (i) and pseudocolor images for the illumination of LCP (ii), RCP (iii), and linearly polarized laser light (iv). The direction of linear polarization is perpendicular to the long axis of the NW. The LCP and RCP laser light were directionally coupled toward the LO (red box) and RO (yellow box) of the NW, respectively. The linearly polarized laser light was directionally coupled toward the RO (yellow box) of the NW. **c** SEM images of a (D-handed chiral Au NC)@SiO₂-NW structure. **d** Pseudocolor dark-field microscopy image of the (D-handed chiral Au NC)@SiO₂-NW structure (i) and pseudocolor images for the illumination of linearly polarized laser light (ii). The linearly polarized laser light was directionally coupled

toward the LO (red box) of the NW. **e** Normalized intensities of the LO and RO in the (L-handed chiral Au NC)@SiO₂-NW structures under the excitation of LCP/RCP light. The data were collected and averaged from 3 L-handed structures. **f, g** Normalized intensities of the LO and RO in the (L-/D-handed chiral Au NC)@SiO₂-NW structures (**f**) and directionality of the SPP propagation (**g**) under the excitation of linearly polarized light. The data were collected and averaged from 3 L-/D-handed structures.

The effect of the shell thickness of the (L-handed chiral Au NC)@SiO₂ nanoparticles on the directional SPP propagation of the hybrid structures was further examined through simulations. Under LCP/RCP excitation, a 15-nm SiO₂ shell leads to a reversed photonic spin-Hall effect, where SPPs propagate along the $-x$ direction for an LCP dipole and $+x$ direction for an RCP dipole (Supplementary Fig. 24a). Although the SiO₂ shell preserves the chiroptical response of the chiral Au NC core, it modifies the near-field coupling of the nanoparticle to the NW. The propagating directions for LCP/RCP exciting light are reversed when the chiral NC is coated with a 15-nm SiO₂ shell. The simulations showed that tuning the shell thickness can also switch the SPP direction for linearly polarized dipoles. Reducing the SiO₂ thickness from 10 nm to 5 nm results in a reversal of the SPP propagation direction, from $+x$ to $-x$ (Supplementary Fig. 24b).

Supplementary Fig. 24 | Simulations of the electric field of the (L-handed chiral Au NC)@SiO₂-(Ag NW) structures with different shell thicknesses. a Distributions of the electric field enhancement in an x - y plane monitor located at the bottom surface of the Ag NW in the (L-handed chiral Au NC)@15-nm-SiO₂-(Ag NW) structure under the excitation of circularly and linearly polarized dipole sources. SPPs propagate along the $-x$ direction for the LCP dipole, whereas SPPs propagate along the $+x$ direction for the RCP dipole. **b** Distributions of the electric field enhancement for the (L-handed chiral Au NC)@SiO₂-(Ag NW) structures with different shell thicknesses under the excitation of linearly polarized dipole sources. The SiO₂ thicknesses are 10 nm, 5 nm, and 2 nm, respectively. SPPs propagate preferentially along the $+x$ direction when the shell thickness is 10 nm or 15 nm. Reducing the SiO₂ thickness from 10 nm to 5 nm results in a reversal of the SPP propagation direction, from $+x$ to $-x$.

We have added discussion on the effect of the distance between the chiral Au NC and the NW to the end of the paragraph right before the section of “Chirality-dependent routing of the valley excitons in WS₂ monolayers”.

“..... The SPPs in the NW can therefore be routed under the excitation of linearly polarized light depending on the chiroptical properties of the attached chiral nanoparticle. To investigate the effect of the distance between the chiral Au NC and the Ag NW on the SPP propagation, the chiral Au NCs were coated with a 15-nm-thick SiO₂ shell (Supplementary Fig. 23). The SiO₂ shell significantly modulated the plasmon coupling and thus reversed the directionality compared to the results of the (chiral Au NC)–NW structures. The simulation results reveal that tuning the SiO₂ thickness (e.g., from 10 nm to 5 nm) can switch the SPP propagation direction under linearly polarized excitation (Supplementary Fig. 24). The SPP routing depends not only on the chiroptical property of the nanoparticle but also on its plasmon coupling with the NW.”

The preparation of the (chiral Au NC)@SiO₂ nanoparticles has been added into Methods.

Preparation of the (chiral Au NC)@SiO₂ nanoparticles

The (chiral Au NC)@SiO₂ core@shell nanoparticles were synthesized by the growth of a mesostructured SiO₂ shell onto the chiral Au NCs through the hydrolysis of tetraethyl orthosilicate (TEOS). The chiral Au NC sample was washed to remove extra CTAB and redispersed in water (1 mL). NaOH (0.1 M, 10 μL) and CTAB (0.1 M, 10 μL) solutions were then added. A SiO₂ precursor was prepared by mixing TEOS (1 mL) with ethanol (9 mL). Under continuous stirring, the precursor solution (10 μL) was added dropwise to the chiral Au NC dispersion every 20 min. After the addition of the precursor solution in 1 h (a total of 30 μL of the precursor solution added), stirring was continued for 5 h to allow for the hydrolysis and formation of a mesostructured SiO₂ shell. The obtained nanoparticle sample was finally washed twice and redispersed in water.

(2) To clarify the effect of the distance between the plasmonic nanostructure and the WS₂ monolayer on the exciton emission dynamics, we have added related simulations on the pumping and decay rates of the hybrid structure.

The simulation settings were consistent with those in Fig. 5, with the only modification being the z -positions of the monitor, dipole sources, WS₂ monolayer, and substrate to control the distance d between the bottom surface of the (chiral Au NC)–(Ag NW) structure and the WS₂ monolayer. The results indicate that the directionality arising from the chirality-dependent SPP coupling remains robust across different d of 6–10 nm. The simulated pumping and decay rates (P and Γ) of the structures with different d are shown in Fig. R1.

The LCP and RCP components of the electric field (P_{LCP} and P_{RCP}) for the (L-handed chiral Au NC)–NW structure under 594 nm linearly polarized excitation are responsible for the pumping rates of excitons from the +K and –K valleys. The asymmetric distributions of both P_{LCP} and P_{RCP} remain consistent for the cases of $d = 6$ nm, 8 nm, and 10 nm, whereas their intensities slightly decrease with increasing d (Fig. R1a).

The decay rates of LCP and RCP dipoles (Γ_{LCP} and Γ_{RCP}) were also simulated to estimate the metal-induced nonradiative damping with increasing d (Fig. R1b–e). The circularly polarized dipole was located at four locations in the x - y plane (xy -1, 2, 3, 4, indicated by the white dots in Fig. R1a) and at different z positions corresponding to d values ranging from 5 nm to 14 nm. The values of Γ_{LCP} and Γ_{RCP} increase with increasing d at all xy positions, whereas the relative

intensity ratio between Γ_{LCP} and Γ_{RCP} remains unchanged. This trend indicates a reduction in non-radiative quenching and the preservation of the chiral asymmetry as d is increased.

Fig. R1 | Simulated pumping and decay rates of the (L-handed chiral Au NC)-(Ag NW)-on-WS₂ structure for different distances. **a** Distributions of P_{LCP} and P_{RCP} in the x - y plane for different distances d of 6 nm, 8 nm, and 10 nm under the excitation of the 594 nm linearly polarized Gaussian beam. d is the distance between the bottom surface of the (L-handed chiral Au NC)-(Ag NW) structure and the WS₂ monolayer. **b–e** Γ_{LCP} and Γ_{RCP} for different d values under the excitation of LCP and RCP dipoles placed at varying positions of 1 (**b**), 2 (**c**), 3 (**d**), and 4 (**e**) in the x - y plane, respectively. The locations of the dipoles in the x - y plane are marked in (**a**).

To evaluate the effect of the plasmonic proximity on WS₂ excitonic emissions, we introduced a 6-nm-thick Al₂O₃ spacer between the Ag NW and the WS₂ monolayer (Fig. R2). When a 514 nm laser was focused either on the lower or upper side of the NW, directional propagation remained absent under linearly polarized excitation, indicating no pronounced asymmetric excitonic emissions induced by the Ag NW under these conditions.

Fig. R2 | PL spectra of the outputs in the (Ag NW)-on-(WS₂ monolayer/6-nm-Al₂O₃) structure under the excitation of linearly polarized laser light. a, b Pseudocolor image (a) and PL spectra of the LO and RO (b) for the illumination of linearly polarized laser light at the lower position of the NW. **c, d** Pseudocolor image (c) and PL spectra of the LO and RO (d) for the illumination of linearly polarized laser light at the upper position of the NW.

Response to Reviewer #2

Comments: Directional light splitting based on the photonic spin-Hall effect is demanding for the development of spin-dependent optical elements but still greatly missing. Here we report on the routing of surface plasmon polaritons (SPPs) by use of chiral Au nanocubes (NCs). Experiments and theoretical modeling have been performed to understand the photonic spin-Hall effect in an Ag nanowire (NW) under excitation of circularly polarized light. The SPPs in the Ag NW can be modulated by the attached chiral Au NCs with opposite chiralities under excitation of linearly polarized light. Hybrid structures are further constructed out of (chiral Au NC)-(Ag NW) assemblies and transition metal dichalcogenide monolayers to selectively induce directional emissions of valley-polarized excitons and enhance the degree of valley polarization. The research group has good experimental equipment and proficient operational skills. Many drawing details expressions are need modified.

Response: We thank the reviewer for their positive assessment of the significance of our work and for acknowledging the experimental effort.

Question 1: In the caption of Figure 1 (e), directly using CPL to describe the experimental design lacks guidance. It is suggested to make the following two modifications:

- 1) supplement the previous text to clarify that CPL has two chirality, namely LCP/RCP.
- 2) clarify in the relevant discussion of Figure 1 (e) that the experiment was conducted separately by LCP and RCP.

Response: We thank the reviewer for the valuable suggestions. We have clarified the description of CPL in the paragraph right after Fig. 1 and in the third paragraph after Fig. 3 for the relevant discussion of Fig. 1e.

“The chiral Au NC in a (chiral Au NC)–(Ag NW) structure was illuminated by a **LCP or RCP** laser beam at the wavelength of 633 nm (Fig. 1c). The scattered laser light at

“..... The normalized intensities of the LO and RO collected from three representative (D-/L-handed chiral Au NC)–NW devices **excited separately by LCP and RCP laser light** are summarized in Fig. 1e. The selected regions of the LO and RO in

***Question 2:** Taking Figure 2 (b) as an example, it is recommended to use two different colors, red and blue, to distinguish between LCP and RCP light sources with different chirality.*

Response: We thank the reviewer for this valuable suggestion. The color of RCP light in Fig. 2b has been changed to blue.

Question 3: The article does not discuss the chirality of gold nanocubes. How is their chirality determined through SEM images? I suggest the author can clearly indicate or supplement the synthesis pathway, or provide corresponding chiral optical effects that can be distinguished. It is recommended that the author refer to the judgment methods of relevant research on the synthesis of this particle for discussion and supplementation.

Response: We thank the reviewer for these helpful suggestions. The chirality of the chiral Au NCs was determined by their morphological handedness observed in their SEM images and extinction asymmetry factor (g -factor).

The SEM images of the chiral Au NCs in Supplementary Fig. 2b, c exhibit a distinct twisted-arm structure with fourfold rotational symmetry. The curvature direction of a half-ring along the arm, clockwise or counterclockwise, can be assigned as D- or L-handedness, respectively (Supplementary Fig. 2b). The D- and L-handed Au NC samples were synthesized through seed-mediated overgrowth on Au nano-octahedra in the presence of chiral glutathione (GSH) molecules, which directed asymmetric growth on high-Miller-index facets. The detailed synthesis method can be found in Methods. The resultant handedness of the chiral nanoparticles was confirmed by circular dichroism (CD) spectroscopy. The aqueous dispersions of the D- and L-handed chiral Au NCs exhibit opposite CD signals across the wavelength range of 500–800 nm, with a measured ensemble extinction g -factor of up to ~ 0.18 . The extinction g -factor is defined as $g_{\text{ext}} = 2 \times (E_{\text{LCP}} - E_{\text{RCP}})/(E_{\text{LCP}} + E_{\text{RCP}})$, where E_{LCP} and E_{RCP} are the extinction intensities of the aqueous chiral Au NC sample under the excitation of LCP and RCP light, respectively. The aqueous D-handed chiral Au NC sample exhibits a positive extinction g -factor band in 520–650 nm, while the L-handed chiral Au NC sample exhibits a negative extinction g -factor band. The extinction g -factor spectra can be found in Fig. 1c of our published work (*ACS Nano* **2024**, *18*, 383).

The determination of chirality through SEM imaging has been added in Supplementary Fig. 2b.

Supplementary Fig. 2 | (Chiral Au NC)–(Ag NW) structures. **a** Schematic of the assembling process of the (chiral Au NC)–NW structures. **b, c** SEM images of the (D-handed chiral Au NC)–NW (**b**) and (L-handed chiral Au NC)–NW structures (**c**). **The chiral Au NC sample exhibits a distinct twisted-arm structure with fourfold rotational symmetry. The curvature direction of a half-ring along the arm, clockwise or counterclockwise, can be assigned as D- or L-handedness, respectively.**

We have added the synthesis method of the chiral Au NCs into the first paragraph of Results.

“The (chiral Au NC)–NW hybrid waveguides were fabricated by assembling chiral Au NCs with distinct chiral morphologies on as-prepared Ag NWs that were deposited on Si/SiO₂

substrates (Supplementary Figs. 1, 2). D- and L-handed Au NCs were synthesized through seed-mediated overgrowth on Au nano-octahedra in the presence of chiral molecules, which directed asymmetric growth on high-Miller-index facets. The Ag NW with a single attached chiral Au NC can be found on

Question 4: Supporting Figure S8(a) suggest placing the coordinate axis at the front to help readers read and understand.

Response: We thank the reviewer for the useful suggestion. The coordinate axis in Supplementary Fig. 8a (now Supplementary Fig. 11a) has been added.

Question 5: Will this quadrilateral structure have resonance problems like previous studies in Figure S17? Ref: L. Shan, F. Zhang, J. Ren et al. Large Purcell enhancement with nanoscale non-reciprocal photon. transmission in chiral gap-plasmon-emitter systems. *Opt Express*, 2020, 28(23): 33890-33899.

Response: We thank the reviewer for this insightful question. In the previous work (*Opt. Express* 2020, 28, 33890), unidirectional CPL transmission is achieved through switching between two distinct plasmon resonance modes, which is enabled by varying the side length of the coupled plasmonic nanoblock. Such an eigenmode switching leads to a reversal of the propagation direction for the same CPL excitation.

In contrast, the chiral Au NCs used in our study were synthesized with highly uniform sizes. They support a single and stable plasmon resonance mode at our operation wavelength of 633 nm. Switching between two distinct plasmon resonance modes does not occur in our work.

Question 6: *The structure of WS₂ and silver nanowires can also cause directional propagation when incident by CPL. Will linearly polarized light incident in the article also have a similar effect? I hope the author can further discuss this issue. If linearly polarized light directly irradiates the WS₂ line structure without causing directional propagation, it directly proves the optical effect of gold nanostructures; If it will lead to directional propagation, the author needs to supplement the discussion on how to avoid the influence of silver nanowires on the experimental results when designing the experiment (such as removing only gold nanoparticles to see if the propagation direction will be affected).*

Response: We conducted the control experiment by studying the PL propagation in a (Ag NW)-on-WS₂ structure (Supplementary Fig. 29). The 514 nm laser light was focused on the upper side of the NW with a lateral offset of ~150 nm from its center. **Under linear polarization excitation, the PL outputs from the LO and RO of the NW remained symmetric**, regardless of whether the polarization was perpendicular or parallel to the long axis of the NW (Supplementary Fig. 29b, c). This is because linearly polarized light can be decomposed into equal LCP and RCP components, which excite symmetric near-field distributions when no chiral scatterer is present.

Even under LCP/RCP excitation, the (Ag NW)-on-WS₂ structure alone showed negligible directionality under our experimental configuration (Supplementary Fig. 29d, e). As noted in a previous work (*Science* **2018**, 359, 443), a pronounced directional response for a (Ag NW)-on-WS₂ structure requires the excitation spot to be offset by ~500 nm from the NW center. This condition was not met with our measurements. In our experiments, the laser was focused on the interface between the chiral Au NC and the Ag NW. The resultant strong near-field confinement localized the electromagnetic field to the gap region.

The polarization-resolved PL spectra under linear polarization excitation further confirmed minimal DCPs at the outputs (Supplementary Fig. 29f, g).

In summary, these control experiments demonstrate that the chiral Au nanoparticle is essential for achieving the directional PL routing observed in our hybrid structures. The Ag NW or WS₂ monolayer alone, under the same excitation geometry, does not produce directional responses.

Supplementary Fig. 29 | PL spectra and polarization-resolved PL spectra of the outputs in the (Ag NW)-on-(WS₂ monolayer) structure under 514 nm laser excitation. a Pseudocolor image of the structures for the illumination of linearly polarized laser light. The laser was focused on the upper side of NW with a lateral offset of ~150 nm from its center. **b, c** PL spectra collected from the LO and RO under linear polarization excitation, with polarization perpendicular (**b**) and parallel (**c**) to the long axis of the Ag NW. **d, e** PL spectra of the LO and RO for the illumination of LCP (**d**) and RCP (**e**) laser light. **f, g** Polarization-resolved PL spectra of LO (**f**) and RO (**g**) for the illumination of linearly polarized laser light. The laser power in (**f, g**) was increased 30-fold to improve signal-to-noise ratio. The PL outputs from the LO and RO of the NW remained symmetric under linear or circular polarization excitation. The difference between the LCP and RCP components of LO/RO was minimal.

We have added discussion on the PL propagation in achiral structures to the end of the paragraph right before Fig. 4.

“..... The absolute DCP values of the ROs are higher than those of the LOs for both the L-handed and D-handed structures. To further clarify the essential role of chiral plasmonic nanoparticles, we conducted a series of control experiments using a bare (Ag NW)-on-WS₂ structure without any attached nanoparticle and (achiral Au NC)-NW-on-WS₂ structures (Supplementary Figs. 29–31). In all control cases, whether no particle, a single achiral Au NC, or an achiral Au NC dimer was attached, the PL intensities from the LO and RO showed negligible differences under linear polarization excitation, and the DCP remained close to zero. These results confirm that the chiral Au nanoparticle is essential for achieving the observed directional routing of PL. In contrast, achiral hybrids or the NW alone do not support directional output under the same excitation geometry.”

Question 7: *Can we discuss the impact of geometric positions? How to control the relative position of gold and silver in the experiment? If the experiment cannot be controlled, should we consider adding relevant simulations for theoretical calculations?*

Response: We thank the reviewer for raising this important point regarding the control and impact of the nanoparticle–NW distance. This question was also raised by Reviewer #1 in Question 5. To address this point, we have performed additional experiments and simulations. The detailed descriptions of new Supplementary Figs. 23, 24 can be found on pages 11–14 in this response letter. We introduced a 15-nm-thick SiO₂ shell onto the surface of the chiral Au NCs to precisely control the spacing, and studied the resultant SPP routing in the (chiral Au NC)@SiO₂-NW structures (Supplementary Fig. 23). The results show that the output directionality under linearly polarized excitation can be reversed compared to the bare (chiral Au NC)-NW structures with the same handedness, confirming that the nanoparticle–NW distance strongly affects the plasmon coupling. We have further conducted simulations on the SPP propagation of the (chiral Au NC)@SiO₂-NW structures with varying SiO₂ shell thicknesses (Supplementary Fig. 24). Tuning the shell thickness of the attached (chiral Au NC)@SiO₂ nanoparticles can switch the preferred SPP propagation direction.

We have added the related discussion of these findings to the end of the paragraph right before the section of “Chirality-dependent routing of the valley excitons in WS₂ monolayers”.

“..... The SPPs in the NW can therefore be routed under the excitation of linearly polarized light depending on the chiroptical properties of the attached chiral nanoparticle. To investigate the effect of the distance between the chiral Au NC and the Ag NW on the SPP propagation, the chiral Au NCs were coated with a 15-nm-thick SiO₂ shell (Supplementary Fig. 23). The SiO₂ shell significantly modulated the plasmon coupling and thus reversed the directionality compared to the results of the (chiral Au NC)-NW structures. The simulation results reveal that tuning the SiO₂ thickness (e.g., from 10 nm to 5 nm) can switch the SPP propagation direction under linearly polarized excitation (Supplementary Fig. 24). The SPP routing depends not only on the chiroptical property of the nanoparticle but also on its plasmon coupling with the NW.”

Question 8: *In this study, the optical properties of gold nanocubes were limited to the conversion of linearly polarized light. The article also mentioned the enhancement effect of the*

structure's composition on electromagnetic fields. Can Raman and other optical properties be further studied?

Response: We thank the reviewer for raising this interesting point. We have verified the broad applicability of chirality-dependent routing using R640 molecules coupled to the (chiral Au NC)–(Ag NW) structures (Supplementary Fig. 33). The chiral Au NCs were coated with R640 and assembled onto the surface of the Ag NW (Supplementary Fig. 33a, d). Under linearly polarized excitation, the PL from R640 exhibited directional SPP propagation that depended on the handedness of the attached chiral Au NC. The D-handed structure routed the PL preferentially toward the LO, whereas the L-handed structure routed it toward the RO (Supplementary Fig. 33b, e). The normalized intensity and the directionality of the outputs, collected from 3 devices for each handedness, showed opposite propagating directions for the D- and L-handed structures (Supplementary Fig. 33c, f, g, h).

Supplementary Fig. 33 | Directional PL from the R640-coupled (chiral Au NC)–NW structures under the excitation of linearly polarized laser light. a, b SEM images of a (D-handed chiral Au NC)@R640–(Ag NW) structure (a) and PL spectra of the LO and RO (b). The direction of linear polarization of the laser light is perpendicular to the long axis of the Ag NW. The fluorophore molecules were embedded in a mesostructured SiO₂ shell surrounding

the chiral Au NC. **c** Directionality of the SPP propagation of 3 (D-handed chiral Au NC)@R640-(Ag NW) structures. **d, e** SEM images of a (L-handed chiral Au NC)@R640-(Ag NW) structure (**d**) and PL spectra of the LO and RO (**e**). **f** Directionality of the SPP propagation of 3 (L-handed chiral Au NC)@R640-(Ag NW) structures. **g, h** Normalized intensities of the LO and RO in the (D-/L-handed chiral Au NC)-NW structures (**g**) and averaged directionality of the PL SPP propagation (**h**). The data were collected and averaged from 3 (L-/D-handed chiral Au NC)-NW structures at the wavelength of 606 nm.

To further validate spin-dependent routing in the fluorescent system, we also examined the R640-coupled chiral structures under circular polarization excitation (Supplementary Fig. 34). Under LCP/RCP excitation, the D-handed structure exhibited PL propagation toward the LO, with a more pronounced difference between the LO and RO under LCP than under RCP illumination (Supplementary Fig. 34a, b). In contrast, the L-handed structure showed reversed propagation and a greater output asymmetry under RCP than under LCP illumination (Supplementary Fig. 34c, d). This enantiomer-specific response confirms that the chiral plasmon-emitter coupling also imposes spin-selective fluorescent emissions and directional propagation. The chirality-dependent photonic spin-Hall effect can effectively direct not only excitonic emissions in WS₂ monolayer but also fluorescence from molecular emitters, supporting the generalizability of the observed chirality-dependent routing.

Supplementary Fig. 34 | PL spectra of the outputs in the (D-/L-handed chiral Au NC)@R640-NW structures under the excitation of circularly polarized laser light. a, b PL spectra of the LO and RO of a (D-handed chiral Au NC)@R640-(Ag NW) structure for the illumination of LCP (**a**) and RCP (**b**) laser light. The D-handed structure exhibits PL propagation toward the LO. The difference between the LO and RO under LCP excitation is larger than that under RCP excitation. **c, d** PL spectra of the LO and RO of a (L-handed chiral Au NC)@R640-(Ag NW) structure for the illumination of LCP (**c**) and RCP (**d**) laser light.

The L-handed structure exhibits PL propagation toward the RO. The difference between the LO and RO under RCP excitation is larger than that under LCP excitation.

We have added the experimental results in the paragraph right before Discussion.

“As an extended demonstration of chirality-dependent routing, we embedded fluorophore molecules R640 in the SiO₂ shell surrounding the chiral Au NC and studied the PL SPP routing of R640 molecules (Supplementary Figs. 33, 34). Under linearly polarized excitation, the R640 emissions exhibit directional SPP propagation that reverses with the handedness of the attached chiral NC. The D-handed structures route the fluorescence preferentially toward the LO, whereas the L-handed structures route it toward the RO. Under LCP/RCP excitation, the D-handed structures exhibit a more pronounced difference between the LO and RO under LCP than under RCP illumination, while the L-handed structures show a greater output asymmetry under RCP than under LCP illumination. These results confirm that the chiral plasmon–emitter coupling imposes a spin-selective directionality not only on excitonic emissions but also on molecular fluorescence emissions, underscoring the generality of chirality-dependent routing.”

The preparation of the (chiral Au NC)–fluorophore nanoparticles have been included in Methods.

Preparation of the (chiral Au NC)–fluorophore nanoparticles

The (Chiral Au NC)–fluorophore nanoparticles were synthesized with a modified procedure to incorporate the R640 fluorophore into the mesostructured SiO₂ shell. The process began identically, with the washing and redispersion of the chiral Au NCs and the addition of the same NaOH and CTAB solutions. The key difference was in the preparation of the precursor solution. R640 (50 mg) was first dissolved in ethanol (5 mL), and the resultant solution was then mixed with TEOS (1 mL) and ethanol (4 mL) to form a fluorophore-containing SiO₂ precursor. The same protocol (3 μL every 20 min for 1 h) was then executed, which was followed by stirring for a total time of 5 h. The SiO₂ shell formed on the surface of the chiral Au NCs with R640 molecules embedded. The final nanoparticles were washed twice and redispersed in water.

Question 9: *figure 1 D-hand and L-hand's color bar have different value, why is the same value?*

Response: We thank the reviewer for raising this question. The identical ranges of values in Fig. 1e for both D- and L-handed structures arise because the images display the normalized intensities, rather than the raw absolute signals acquired by the camera.

The normalization procedure is as follows. The photon counts from the LO and RO were first obtained by a fitting process as shown in Supplementary Fig. 4 and then divided by their acquisition time durations to obtain the intensities of I_{LO} and I_{RO} . We then calculated the normalized intensity for each output. The normalized I_{LO} equals $I_{LO}/(I_{LO} + I_{RO})$. The normalized I_{RO} equals $I_{RO}/(I_{LO} + I_{RO})$. The same range of 0–1 of the normalized intensities can therefore be used to compare the contrast between the LO and RO.

Journal: *Nature Communications*

Manuscript ID: NCOMMS-25-63656A

Title: “Photonic spin-Hall effect in chiral plasmonic assemblies”

Authors: Yilin Chen, Yang Chen, Yini Fang, Ruoqi Ai, Ximin Cui, Xiaolu Zhuo, Jianfang Wang

Response to Reviewer #1

Comments: The revised version has addressed the comments, and the manuscript has been improved accordingly. I think it could be accepted for publication now.

Response: We thank this reviewer for the effort on evaluating our work again, the highly positive comment, and the great help in the improvement of our manuscript.

Response to Reviewer #2

Comments: Directional light splitting relying on the photonic spin-Hall effect is crucial for spin-dependent optical elements yet remains largely unexplored, and this work reports the routing of surface plasmon polaritons (SPPs) using chiral Au nanocubes (NCs). Experiments and theoretical modeling are conducted to investigate the photonic spin-Hall effect in Ag nanowires (NWs) under circularly polarized light excitation, and linearly polarized light-driven SPPs in Ag NWs can be modulated by attached chiral Au NCs with opposite chiralities. Hybrid structures composed of chiral Au NC-Ag NW assemblies and transition metal dichalcogenide monolayers are constructed to selectively trigger directional emissions of valley-polarized excitons and boost valley polarization degree, whose routing mechanism is clarified via numerical simulations and holds great significance for valleytronic circuit development.

The quality of the data is technically sound and presented in sufficient detail. The level of support for the conclusions are sufficiently strong evidence is provided for the authors' claims and all appropriate controls have been included. These results will be important to the field and advance understanding in a way that will move the field forward.

Response: We thank this reviewer for the effort on evaluating our work again, the highly positive comment, and the great help in the improvement of our manuscript.